# Ligand effect on switching the rate-determining step of water oxidation in atomically precise metal nanoclusters

Zhihe Liu[1,2], Hua Tan[3], Bo Li[4], Zehua Hu[3], De-en Jiang [4], Qiaofeng Yao [1] ✉, Lei Wang [2] ✉ & Jianping Xie [1,2] ✉

The ligand effects of atomically precise metal nanoclusters on electrocatalysis kinetics have been rarely revealed. Herein, we employ atomically precise $Au_{25}$ nanoclusters with different ligands (i.e., para-mercaptobenzoic acid, 6-mercaptohexanoic acid, and homocysteine) as paradigm electrocatalysts to demonstrate oxygen evolution reaction rate-determining step switching through ligand engineering. $Au_{25}$ nanoclusters capped by para-mercaptobenzoic acid exhibit a better performance with nearly 4 times higher than that of $Au_{25}$ NCs capped by other two ligands. We deduce that para-mercaptobenzoic acid with a stronger electron-withdrawing ability establishes more partial positive charges on Au(I) (i.e., active sites) for facilitating feasible adsorption of $OH^-$ in alkaline media. X-ray photo-electron spectroscopy and theoretical study indicate a profound electron transfer from Au(I) to para-mercaptobenzoic acid. The Tafel slope and in situ Raman spectroscopy suggest different ligands trigger different rate-determining step for these $Au_{25}$ nanoclusters. The mechanistic insights reported here can add to the acceptance of atomically precise metal nanoclusters as effective electrocatalysts.

Atomically precise thiolate-protected noble metal nanoclusters (NCs) possessing an ultrasmall core size (<3 nm) have emerged as a new family of metal NPs for effective electrocatalysis. They have atomically precise structure and composition, which is descriptive by a molecular-like formula of $[M_n(SR)_m]^q$ (n, m and q are the numbers of metal atoms, thiolate ligands (SR), and net charge of individual cluster, respectively)[1–3]. More importantly, the atomically precise composition and structure render them as a good platform to study the structure-performance relationship of electrocatalysts at the unprecedented molecular and atomic levels. Due to the strong quantum confinement effects in this ultra-small size regime, metal NCs display molecular-like numerous properties including quantized charging, discrete energy levels, and redox behaviors[4–8]. Thanks to these unique physiochemical properties and atomically precise structures, metal NCs have been

widely applied in diverse electrocatalytic reactions to illustrate underlying structure-property relationships[9]. Therefore, metal NCs have risen as good candidates of model electrocatalysts in both cluster chemistry and applied electrocatalysis research[10,11].

The electrocatalysis activity and stability highly rely on the structures of metal NCs[12]. The physiochemical property of metal NCs can therefore be tailored by rationally engineering the following three domains of structures: i) metal core, ii) metal-thiolate interface between the metal core and protecting ligands, and iii) protecting ligands. Generally, it has become a well-accepted consensus that the catalytic performance of metal NCs can be tuned by rationally tailoring the above structural attributes and thus electronic structures[13]. For instance, much progress has been made on the fine tuning of the metal core in terms of their composition and packing structure, largely

[1]Joint School of National University of Singapore and Tianjin University International Campus of Tianjin University Binhai New City Fuzhou, Fuzhou 350207, PR China. [2]Department of Chemical and Biomolecular Engineering National University of, Singapore 117585, Singapore. [3]Division of Physics and Applied Physics, School of Physical and Mathematical Sciences Nanyang Technological University, Singapore 637371, Singapore. [4]Department of Chemical and Biomolecular Engineering, Vanderbilt University, Nashville, TN 37235, USA. ✉e-mail: qfyao@tjufz.org.cn; wanglei8@nus.edu.sg; chexiej@nus.edu.sg

boosting the electrocatalytic activity of metal NCs[14,15]. The change in superatomic electronic configuration and the follow-up positively-shifted reduction potentials account for the activity enhancement. Besides metal core, protecting ligands can also significantly influence the electrocatalytic performance of metal NCs[16,17]. In particular, ligands work as outmost layer of metal NCs, directly interacting with the reaction environment (e.g., electrolyte, solvent, and reactant ions/molecules). Moreover, they determine the electronic structures of metal NCs through the atomic orbitals coupling between the anchoring atoms (e.g., sulfur) and metal atoms[17–19]. Considerable research efforts have been devoted to revealing the ligand effects on the electronic structures in terms of the change of HOMO-LUMO gap, which affects the adsorption on reactants or reaction intermediates[17,20]. Nevertheless, an understanding toward the ligand effect on the electrocatalysis rate-determining step (RDS) is currently lacking at the molecular and atomic levels. This should be largely attributed to the intrinsic difficulty in capturing the key reaction intermediates in the electrocatalytic reactions in the presence of $[Au_{25}(SR)_{18}]^-$ NCs capped by varied SR ligands, although the identification of the key intermediates is crucial to understand the electrocatalytic pathways with different one-step reaction kinetics.

Herein, we implement $[Au_{25}(SR)_{18}]^-$ NCs separately capped by three different thiolate ligands with similar size (approximated by their similar molecule weights) but different structures (i.e., para-mercaptobenzoic acid (pMBA), 6-mercaptohexanoic acid (MHA), and homocysteine (HCys) as model NCs (Fig. 1a), and study the ligand effect on electrocatalytic OER kinetics (e.g., rate-determining step) of $[Au_{25}(SR)_{18}]^-$ NCs. It has been increasingly known that the Au(I) in the protecting shell of $[Au_n(SR)_m]^q$ NCs is the active sites for feasible adsorption of reactants/intermediates in diverse catalytic reactions[21]. We therefore hypothesize that SR ligands with similar size, but varied electron-withdrawing capability should be able to induce varied electronegativity of Au(I) in $[Au_{25}(SR)_{18}]^-$ NCs. The diversity in electronegativity of Au(I) can thus lead to varied impacts on adsorption of the involved reactants with negative charges. Our systemic study reveals that the varied electron-withdrawing capability of pMBA, MHA and HCys can induce distinct changes in the RDS of OER. Specifically, Au NCs capped by ligands with stronger electron-withdrawing ability could develop more local partial positive charge of Au(I) in the Au(I)-SR motif of NCs, facilitating feasible adsorption of OH⁻ on the active Au(I) sites in the alkaline media[22]. Such favorable adsorption of OH⁻ can lower the activation energy barrier for intermediates, and thereby altering the RDS of OER[23].

## Results

### Au₂₅ NCs synthesis and characterization

$[Au_{25}(SR)_{18}]^-$ was chosen as model cluster in this study. This is because of its good stability, synthesis feasibility, and known crystal structure[24–26]. As shown in Fig. 1a, $[Au_{25}(SR)_{18}]^-$ NCs possess a Au(0)@Au(I)-SR core-shell structure with quasi-$D_{2h}$ symmetry, where a 13-atom Au(0) core is capped by six staple-like SR-[Au(I)-SR]₂ motifs[27–29]. The development of the formal positive charge on the Au atoms in the protecting shell should be largely attributed to the electron-withdrawing capability of protecting SR ligands. Therefore, deliberate control over electron-withdrawing capability of R moiety in SR ligands can offer a good means for fine tuning the electronegativity of Au(I) in the protecting shell. Three representative water-soluble ligands with similar molecular weight ($M_w = 154$ for pMBA, $M_w = 135$ for HCys, and $M_w = 148$ for MHA) but different electron withdrawing ability (pMBA > HCys > MHA) are selected, to exemplify the ligand effects on the OER performance of $[Au_{25}(SR)_{18}]^-$ NCs. The p-π electron delocalization between benzene ring and sulfur atom renders pMBA with stronger electron-withdrawing ability than the other two ligands[30]. HCys has stronger electron-withdrawing ability than that of MHA, which is interpreted by the XPS spectra of S 2p in the Supplementary Fig. 1. The positive shift of S 2p from MHA to HCys, to pMBA manifests the electron-withdrawing capability of sulfur in ligands[31].

The Au₂₅ NCs were synthesized based on a mild-reduction strategy reported by our group[32,33]. The detailed synthetic procedures are provided in the Supplementary Information. These as-synthesized Au

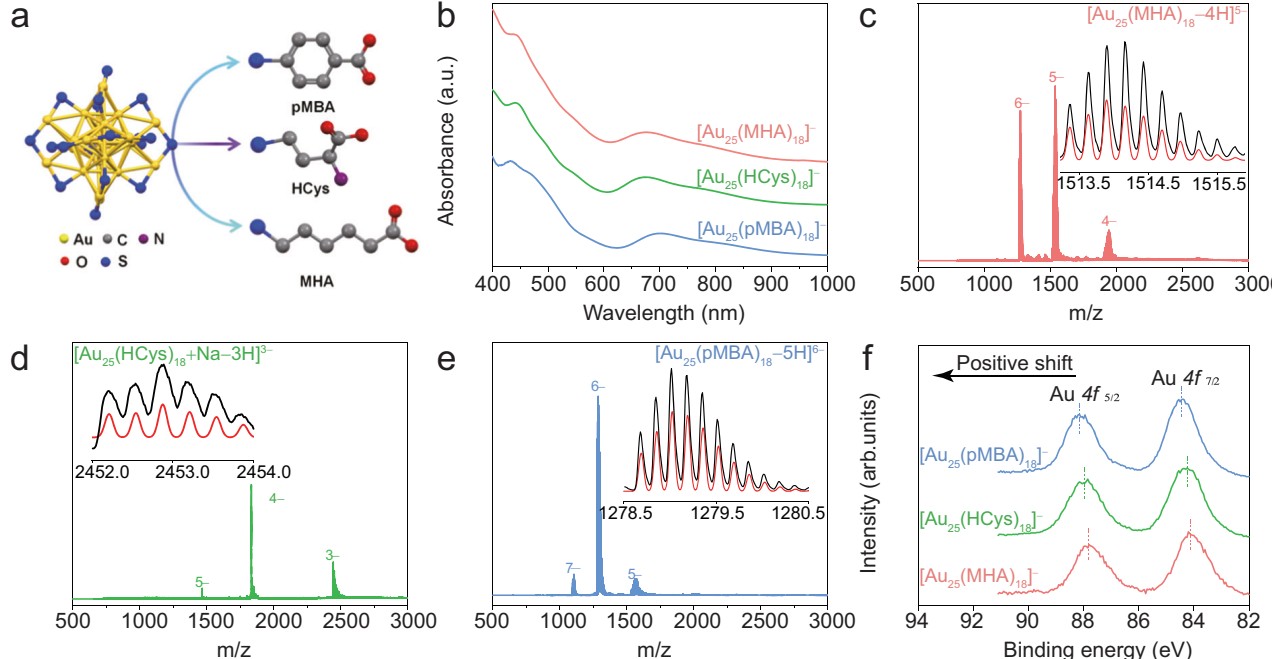

**Fig. 1 | Molecular and structural characterization of synthesized Au₂₅ NCs.**
**a** Schematic illustration of Au₂₅ NCs capped by different ligands.
**b** Ultraviolet−visible (UV−Vis) absorption. **c**−**e** Electrospray ionization mass spectrometry (ESI-MS, in negative-ion mode), and **f** High-resolution Au 4f XPS spectra of $[Au_{25}(SR)_{18}]^-$ NCs where SR are MHA, HCys, and pMBA, respectively. The insets in **c**−**e** are isotope patterns of corresponding cluster ions verifying the accuracy of mass spectrum assignment.

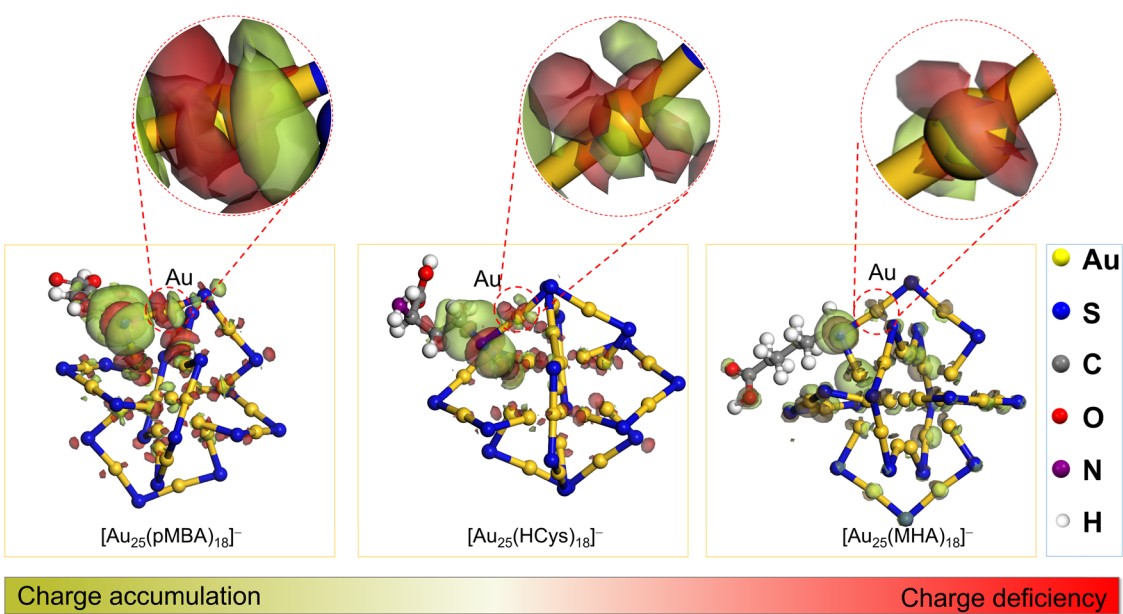

**Fig. 2 | Differential charge density maps of Au$_{25}$ NCs capped by different ligands.** Only one thiolate ligand is shown in the structural model of individual cluster for clarity purpose. A charge accumulation/deficiency for sulfur and the correspondingly bonded gold is highlighted.

NCs are reddish brown in aqueous solutions (Supplementary Fig. 2) and display characteristic absorption features of pure [Au$_{25}$(SR)$_{18}$]$^-$ NCs in their ultraviolet–visible (UV–Vis) absorption spectra (Fig. 1b). Despite similar UV–vis absorption profiles, the detailed positions of characteristic peaks exhibit notable shift for Au$_{25}$ NCs capped by different SR ligands: [Au$_{25}$(MHA)$_{18}$]$^-$ (440, 552, 670, and 760 nm), [Au$_{25}$(HCys)$_{18}$]$^-$ (443, 548, 670 and 801 nm), and [Au$_{25}$(pMBA)$_{18}$]$^-$ (460, 575, 690 and 815 nm). The slight variation for the absorption peak positions could be attributed to the altered HOMO-LUMO transitions caused by different ligands. Due to the combined delocalization and induction effects, the LUMO level was lowered with the increasing electron-withdrawing capability of ligands[34]. The precise molecular formulae of the synthesized Au NCs were further confirmed to be [Au$_{25}$(SR)$_{18}$]$^-$ by electrospray ionization mass spectrometry (ESI-MS). Taking the ESI-MS spectrum of [Au$_{25}$(MHA)$_{18}$]$^-$ NCs as an example (Fig. 1c), there are three sets of peaks observed in a broad range of m/z = 500–4000, which account for Au$_{25}$ NCs capped by MHA carrying 6, 5, and 4 negative charges, respectively. The experimental (black) and simulated (red) isotope patterns of [Au$_{25}$(MHA)$_{18}$−5H]$^{5-}$ show excellent match (inset of Fig. 1c), exemplifying the good accuracy of our assignment of mass spectrum. Similarly, all peaks in ESI-MS spectra of [Au$_{25}$(HCys)$_{18}$]$^-$ (Fig. 1d) and [Au$_{25}$(pMBA)$_{18}$]$^-$(Fig. 1e) can be assigned to their cluster peaks carrying varied charges (e.g., m/z = 1470 (5 e$^-$), 1840 (4 e$^-$), and 2460 (3 e$^-$) for [Au$_{25}$(HCys)$_{18}$]$^-$; and m/z = 1100 (6 e$^-$), 1290 (5 e$^-$), and 1570 (4 e$^-$) for [Au$_{25}$(pMBA)$_{18}$]$^-$). All three Au NCs obtained in this study show ultrasmall core size of ~1 nm based on transmission electron microscopy (TEM) images (Supplementary Figs. 3–5), which are in good agreement with those measured by X-ray crystallography[34].

In order to shed light on the ligand effects on the electronic structures of Au(I) in Au NCs, we carried out X-ray photo-electron spectroscopy (XPS) measurement. The high-resolution XPS spectra of Au 4f of Au NCs were depicted in Fig. 1f, which manifests a dramatically positive shift of Au 4f binding energy for [Au$_{25}$(pMBA)$_{18}$]$^-$ NCs in comparison to those of [Au$_{25}$(MHA)$_{18}$]$^-$ and [Au$_{25}$(HCys)$_{18}$]$^-$ NCs. Such increase in binding energy readily means that the more electron-withdrawing pMBA ligands can induce more partial positive charge on Au(I) of [Au$_{25}$(pMBA)$_{18}$]$^-$. Next, we theoretically analyzed the electronic structures of sulfur of ligand and bonded Au(I) in Au$_{25}$ NCs (See

Supplementary Information for more details). The differential charge density maps in Fig. 2 suggest a charge accumulation/deficiency for sulfur and the correspondingly bonded gold (as shown in the dotted red circles). The Au(I) exhibits an obvious charge deficiency in [Au$_{25}$(pMBA)$_{18}$]$^-$ NCs, which is affected by the neighboring sulfur of pMBA. In stark contrast, the charge depletion of Au(I) is relatively weaker for [Au$_{25}$(HCys)$_{18}$]$^-$ and [Au$_{25}$(MHA)$_{18}$]$^-$. Taken together, the ligand with different electron-withdrawing ability renders Au(I) of Au$_{25}$ NCs with partial positive charge of different extent.

## OER performance in alkaline media

The electrocatalytic OER activities of the synthesized Au$_{25}$ NCs were evaluated using a standard three-electrode system in a H-type cell (Supplementary Fig. 6) containing O$_2$-saturated 1.0 M KOH electrolyte. All potentials are reported versus reversible hydrogen electrode (RHE). Representative linear sweep voltammetry (LSV) curves for OER were recorded with a scan rate of 5 mV s$^{-1}$ (Fig. 3a). A typical current density of 10 mA/cm$^2$ was recorded at an overpotential of 360 mV for [Au$_{25}$(pMBA)$_{18}$]$^-$ NCs, which is lower than that of [Au$_{25}$(HCys)$_{18}$]$^-$ NCs (470 mV) and [Au$_{25}$(HCys)$_{18}$ NCs]$^-$ (540 mV). The pristine carbon paper shows minimal anodic current within the same potential range, eliminating the impact of substrate on the improved electrocatalytic performance. Moreover, the LSV curve comparison for [Au$_{25}$(pMBA)$_{18}$]$^-$ NC and Au-pMBA complex suggests the dominant role of hierarchical cluster structure for effective electrocatalysis (Supplementary Fig. 7). To address the merits of discrete electronic structures for Au NCs, we also compared their electrocatalytic performance with that of larger Au nanoparticles capped by pMBA (NPs, ~5 nm in diameter, prepared by NaBH$_4$ reduction (Supplementary Fig. 8). The LSV curves in Supplementary Fig. 9 indicated that Au NPs deliver the highest current density of 2.5 mA/cm$^2$ within the same potential window from 1.1 to 1.8 V. The weak bonding of OH$^-$ on gold surface may be responsible for the unsatisfactory activity for the Au NPs[35]. Moreover, we also compared our Au$_{25}$ NCs with other thiolate-protected transition metal electrocatalysts in terms of the overpotential required for reaching a current density of 10 mA/cm$^2$ (Supplementary Fig. 10). As can be seen, [Au$_{25}$(pMBA)$_{18}$]$^-$ NCs display a better performance, while [Au$_{25}$(HCys)$_{18}$]$^-$ and [Au$_{25}$(MHA)$_{18}$]$^-$ are comparable to other transition metal based electrocatalysts. To further confirm the valence state of

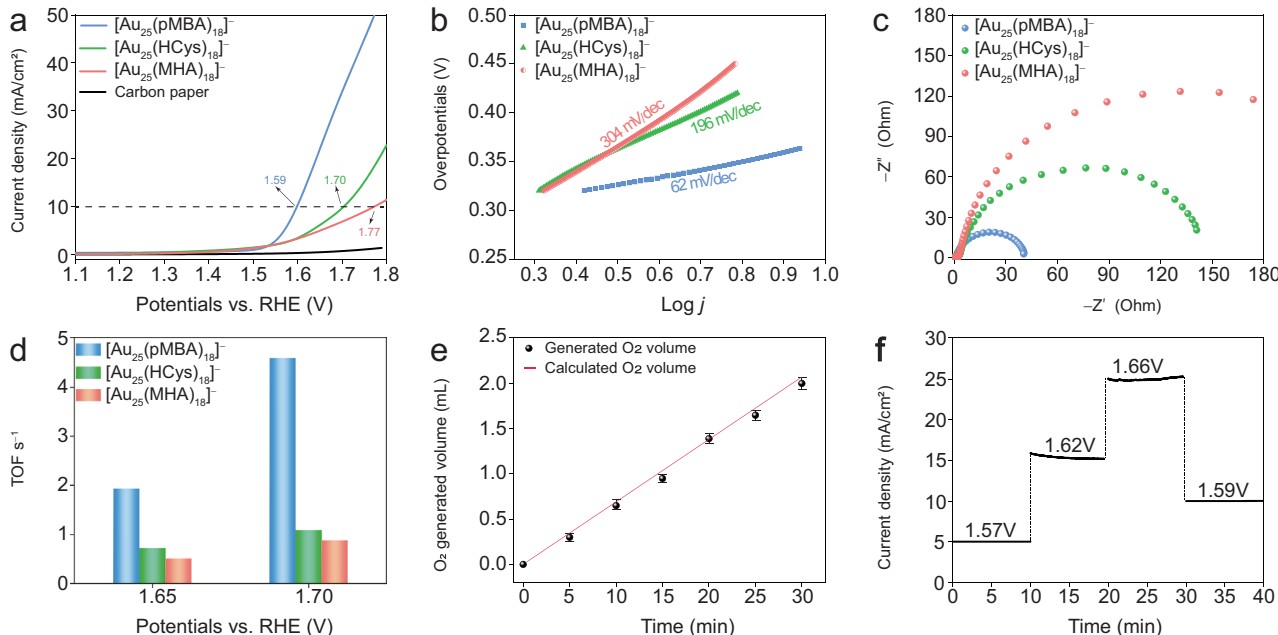

**Fig. 3 | Electrocatalytic performance of different-ligands-capped [Au₂₅(SR)₁₈]⁻ NCs for OER. a** Linear sweep voltammetry (LSV) curves recorded at a scan rate of 5 mV s⁻¹ after iR-corrected. **b** Corresponding Tafel plots. **c** Electrochemical impedance spectroscopy (EIS) Nyquist plots at 1.65 V vs. RHE. **d** Turn-over frequency (TOF) values at 1.65 and 1.7 V of Au₂₅ NCs capped by MHA, HCys, and pMBA. **e** O₂ production volumes as a function of water-splitting time by using [Au₂₅(pMBA)₁₈]⁻ as electrocatalysts: the circles are experimentally obtained O₂ volumes, while the solid line indicates theoretical value of O₂ calculated by assuming a 100% Faradaic efficiency for the anode reaction at the current density of 10 mA/cm² (The error bars correspond to one standard deviation) and **f** I-t curves of [Au₂₅(pMBA)₁₈]⁻ NCs with different applied electrode potentials. The electrocatalytic reactions were carried out in O₂-saturated aqueous solution of 1.0 M KOH.

Au₂₅ NCs before starting OER, cyclic voltammetry (CV) curves were recorded within the same potential windows for LSV (Supplementary Figs. 11–13). The CV curves suggest a change from [Au₂₅(SR)₁₈]⁻ NCs to [Au₂₅(SR)₁₈]⁰ before OER occurrence. Such one-electron transfer behavior[36] is caused by quantized charging effect of monolayer-protected metal NCs (See Supplementary Information for more details), which indicates the capability of accepting electrons for [Au₂₅(SR)₁₈]⁰. This result is in good accordance with those reported in the literatures[37].

To evaluate the OER kinetics, we further investigated the Tafel plot derived from LSV curves. The Tafel equation formulates a mathematical relationship between the current density and the applied potential, which is vital to understand electrochemical kinetics[38]. As shown in Fig. 3b, the Tafel slope of [Au₂₅(pMBA)₁₈]⁻ NCs within the overpotential range from 0.32 to 0.36 V is 62 mV/ dec, suggesting the decomposition of Au-O-OH might be slower than other elementary steps. In comparison, [Au₂₅(MHA)₁₈]⁻ and [Au₂₅(HCys)₁₈]⁻ NCs deliver a higher Tafel slope of 304 and 196 mV/dec, respectively. These values imply that the RDS is dominated by deprotonation on Au-OH[39]. The above-mentioned different Tafel slope values suggest the RDS is changed by varying the protecting ligands from MHA or HCys to pMBA. The larger Tafel slope reflects stronger polarization as the current density rises. Such higher Tafel slope value presumably has a variety of causes. On the one hand, Tafel slope varies depending on different RDS. On the other hand, the higher activation energy on intermediates possibly makes Tafel slope value higher[40]. The double-layer capacitance (C_dl) of Au₂₅ NCs capped by three ligands recorded in non-Faradaic region was unveiled to evaluate the local microenvironment change caused by Au(I) with different contents of partial positive charges. (Supplementary Figs. 14–16). The double layer capacitance can be used to estimate ECSA in electrocatalysis[41]. The C_dl value for [Au₂₅(pMBA)₁₈]⁻ NCs (28.85 mF/cm²) is close to those of [Au₂₅(MHA)₁₈]⁻ (17.30 mF/cm²) and [Au₂₅(HCys)₁₈]⁻ (20.20 mF/cm²), indicating their superior kinetics (Supplementary Fig. 17). Moreover,

electrochemical impedance spectroscopy (EIS) on [Au₂₅(SR)₁₈]⁻ NCs was conducted to infer the charge-transfer resistance. [Au₂₅(pMBA)₁₈]⁻ NCs feature the smallest semi-circular arc, indicating their faster electrode kinetics (Fig. 3c). In addition, compared with traditional turn-over frequency (TOF) calculation involved electrocatalytic surface area and active site numbers in cell unit, the well-defined molecular structure enables Au₂₅ NCs to provide a more reasonable understanding towards the intrinsic electrocatalytic performance due to their precise active site numbers. The TOF value of [Au₂₅(pMBA)₁₈]⁻ NCs is nearly 4 times higher than that of [Au₂₅(MHA)₁₈]⁻ and [Au₂₅(HCys)₁₈]⁻ NCs, further confirming their better intrinsic activity (Fig. 3d). In order to correlate the Au charge deficiency of Au₂₅ NCs to their TOF in OER, we determined the Au(I)/Au(0) ratio in individual Au₂₅ NCs by deconvoluting Au 4f peak in their XPS spectra. Then we plotted TOF vs. Au(I)/Au(0) ratio in Supplementary Fig. 18. As can be seen in Supplementary Fig. 18, an ascending trend is observed for TOF with increasing Au(I)/Au(0) ratio, in good consistency with electron-withdrawing capability of examined thiolate ligands. The volume of generated oxygen in the OER process for [Au₂₅(pMBA)₁₈]⁻ NCs was recorded by online trace gas analysis system-gas chromatography (Fig. 3e). A linear relationship matches well with the computed cumulative charge-volume ratio, demonstrating the cathodic FE of [Au₂₅(pMBA)₁₈]⁻ NCs is as high as 99.5%. The long-term stability of electrocatalysts is also crucial for their practical implementations[42,43]. As shown in Fig. 3f, [Au₂₅(pMBA)₁₈]⁻ NCs exhibit a practically invariant current density within the applied potential window from 1.57 to 1.66 V, suggesting their good electrochemical stability. The strong π-π stacking interactions of pMBA should account for the good structural and thus electrocatalytic stability of corresponding Au₂₅ NCs. By contrast, [Au₂₅(HCys)₁₈]⁻ NCs maintain its current density of 96% in 10 h (Supplementary Fig. 19). The current density of [Au₂₅(MHA)₁₈]⁻ NCs degrade to the 60% of the initial value, which may result from the oxidative degradation of [Au₂₅(MHA)₁₈]⁻ NCs (Supplementary Fig. 20)[44]. In addition, we conducted additional experiment to confirm the structural stability of

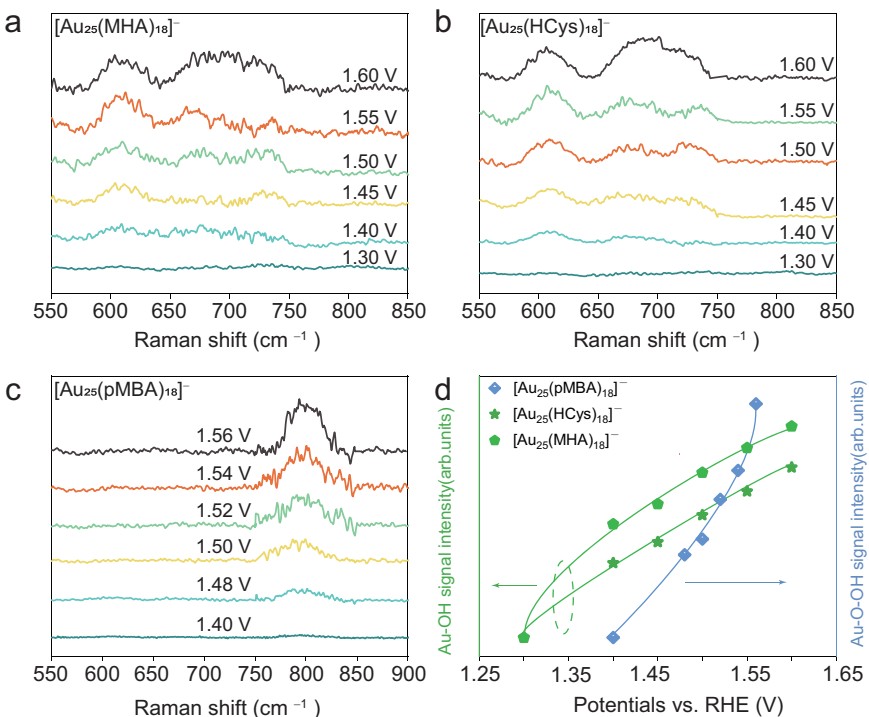

**Fig. 4 | In situ Raman spectra of [Au₂₅(SR)₁₈]⁻ NCs in OER performed in 1.0 M KOH solution.** In situ Raman signal recorded at varied applied potentials for **a** [Au$_{25}$(MHA)$_{18}$]⁻, **b** [Au$_{25}$(HCys)$_{18}$]⁻ and **c** [Au$_{25}$(pMBA)$_{18}$]⁻ NCs. **d** Plot of the Raman intensity vs. the increasing potentials.

Au$_{25}$ NCs during the electrocatalytic reactions. We performed UV–vis absorption spectroscopy and ESI-MS to determine the formula of Au NCs before and after OER. The UV–vis absorption spectra before and after 200 cycles of LSV tests show almost identical profiles with minor differences in the intensity, where slight decrease in absorption intensity was observed at ~695 nm (Supplementary Fig. 21). This reflects that the majority of [Au$_{25}$(pMBA)$_{18}$]⁻ survive after the OER although a trivial degradation is not avoidable. The ESI-MS of Au$_{25}$NCs capped by pMBA after OER process further confirm their unchanged formula after OER (Supplementary Fig. 22). With respect to [Au$_{25}$(MHA)$_{18}$]⁻ and [Au$_{25}$(HCys)$_{18}$]⁻ NCs, more distinct decrease in absorption peak intensity was observed. (Supplementary Figs. 23 and 24), indicating that they are less stable than [Au$_{25}$(pMBA)$_{18}$]⁻. Nevertheless, their formulae are [Au$_{25}$(MHA)$_{18}$]⁰ and [Au$_{25}$(HCys)$_{18}$]⁰ NCs, which are unearthed by ESI-MS (Supplementary Figs. 25 and 26). Overall, the atomic-level morphology is not changed for Au$_{25}$ NCs capped by these three ligands before and after the OER tests.

Capture the intermediates in OER. We next proceed to monitor the key intermediates on the surface of [Au$_{25}$(SR)$_{18}$]⁻ NCs for OER using in situ Raman spectroscopy. A confocal Raman microscope with 532 nm excitation was used to record these spectra (Supplementary Fig. 27). The in situ Raman analysis was carried out at various applied potentials around the onset potential. The Raman signals were recorded in a spectral window from 550 to 850 cm⁻¹, which covers almost all reported oxygen species bonded to Au atom. As shown in Fig. 4a, b, there are no noticeable signals observed on the surface of [Au$_{25}$(MHA)$_{18}$]⁻ and [Au$_{25}$(HCys)$_{18}$]⁻ NCs likely due to the unfavorable adsorption of OH⁻ when the applied potential is below 1.4 V. With the applied potentials increasing to 1.4 V, noticeable Raman signals located at ~600 and 700 cm⁻¹ were observed for [Au$_{25}$(MHA)$_{18}$]⁻ and [Au$_{25}$(HCys)$_{18}$]⁻ NCs. These signals should belong to the Au-OH stretching vibration, which are close to the values of Au-OH at 635 and 677 cm⁻¹ according to the previous study[35]. The deviation of the Raman shift in our Au$_{25}$ NC based electrocatalyst may be attributed to the size effect[45]. In contrast to Raman spectra recorded for

[Au$_{25}$(MHA)$_{18}$]⁻ and [Au$_{25}$(HCys)$_{18}$]⁻ under the same conditions, the Raman spectra of [Au$_{25}$(pMBA)$_{18}$]⁻ NCs shows an emerging peak at ~800 cm⁻¹ with increasing applied potentials (Fig. 4c).The peak at ~800 cm⁻¹ is assignable to Au-O-OH species, since similar peak at 815–830 cm⁻¹ was documented as O-O vibration of Au-O-OH in Au electrode system[35,45]. It has been widely accepted that OER current depends on electrocatalytic reaction rate. The reaction rate is closely associated with the coverage of intermediate species. Therefore, the coverage for the intermediates will be affected by the applied potential, and a threshold value is required for a specific intermediate coverage to trigger its extensive conversion[46]. Specifically, if an intermediate specie participates in the RDS as reactant, it will accumulate on the surface of metal NCs, boosting its surface coverage. This also provide an opportunity for us to capture the intermediates using in situ Raman spectroscopy. Although the peak positions vary with the protecting ligand changed from aromatic (i.e., pMBA) to aliphatic (i.e., MHA and HCys), the peak intensity shows a ubiquitous ascending trend with the increase of the applied potentials regardless of the capping agent used (Fig. 4d), suggesting that the coverage of intermediate species, either Au-OH or Au-O-OH, increases with the elevated overpotential. Combined with Tafel slope analysis on different RDS for Au$_{25}$ NCs capped by different ligands, the Raman data provide molecular-level information on the intermediate species accumulated on the surface of Au$_{25}$ NCs, hinting on the RDS of OER. The above data demonstrate the electronegativity of Au(I) in the motif of Au NCs can be tuned by the ligands with different electron-withdrawing ability. The Au(I) electronegativity variation can result in distinguishing OER behaviors. pMBA with conjugated benzene ring induce a stronger electronegativity of Au(I) in [Au$_{25}$(pMBA)$_{18}$]⁻ NCs, which endows [Au$_{25}$(pMBA)$_{18}$]⁻ NCs with the best OER performance among the synthesized [Au$_{25}$(SR)$_{18}$]⁻ NCs.

Taking the varied key intermediates into consideration, it can be proposed that the RDS of OER on Au NCs is highly related to their capping ligands (Fig. 5). To better understand the ligand effects, we adapt a typical OER pathway in the alkaline medium for [Au$_{25}$(SR)$_{18}$]⁻

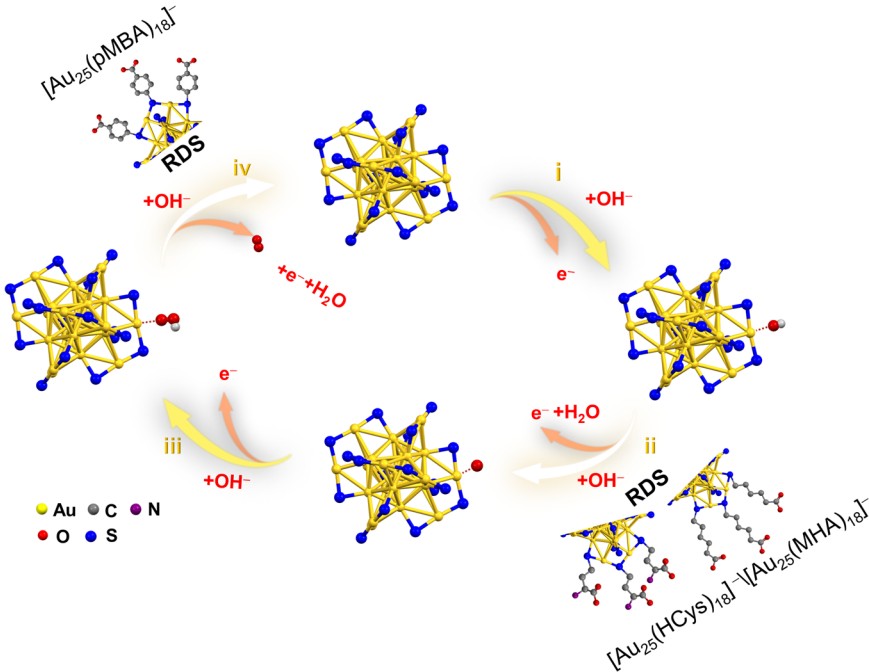

**Fig. 5 | Proposed schematic illustration of OER pathways of [Au₂₅(SR)₁₈]⁻ NCs.** [Au₂₅(SR)₁₈]⁻ have the same OER pathways but different RDS caused by varied ligands.

NCs consisting of four steps: i) OH⁻ ions adsorption on Au(I) sites to form Au-OH bond; ii) deprotonation of the Au-OH species to afford Au-O species; iii) Au-O species coupled with anther OH⁻ to produce Au-O-OH species; and iv) Au-O-O species decomposition to obtain $O_2$ molecule. In the aforementioned catalytic steps, the Au-OH, Au-O, Au-O-OH species are dominant intermediates in electrocatalytic OER reaction[47]. To understand the ligand effect on the OER performance of Au₂₅ NCs in alkaline conditions, the adsorption energies of OH and O on Au₂₅ NCs were calculated to compare pMBA and MHA ligands. To save the computational cost, we employed the models of [Au₂₅(SCH₃)₁₅(SR)₃]³⁻ (SR = pMBA or MHA) where three deprotonated real thiolate ligands are present on the same dimeric motif while other thiolate ligands are simplified as SCH₃ (more computational details can be found in the section Methods). As shown in supplementary Fig. 28 and 29, both O and OH adsorb on an Au(I) atom in the dimeric motif and, more importantly, their adsorption is less stronger with the presence of the pMBA ligand than the MHA ligand, indicating that *OH and *O species are easier to be activated on [Au₂₅(pMBA)₁₈]⁻. Such correlations of adsorption energy with OER activities have been observed by others as well[48,49]. It should also be noted that the OER catalyzed by Au NCs possess very different pathways from those of bulk Au. On the bulk Au surface, it is widely accepted that the adsorbed OH⁻ (Au-OH_ads) will be converted into $Au_2O_3$ via $Au(OH)_3$ before its ultimate decomposition into $O_2$[47,50].

In summary, a series of water-soluble Au₂₅ NCs capped by three typical ligands were employed as a model to highlight the ligand effect on electrocatalytic OER kinetics. The RDS of OER for Au₂₅ NCs are altered by varying MHA and HCys to pMBA. Furthermore, Au₂₅ NCs capped pMBA with stronger electron withdrawing ability exhibit satisfied OER performance, significantly better than the other two ligand protected Au₂₅ NCs. The TOF of [Au₂₅(pMBA₁₈)]⁻ NCs is ~ 4 to 5 times higher than other two Au₂₅ NCs. Mechanistic investigations revealed that ligands with stronger electron-withdrawing capability provoke more positive charges of active sites (i.e., Au(I) species) in Au₂₅ NCs. Such more partial positive charges play a key role in changing the RDS of OER. This work not only provides a molecular-level understanding towards the ligand effects on the electronic structures of active sites of metal NCs and the follow-up

electrocatalytic performance, but also stimulates engineering strategy investigations in the field of cluster chemistry and electrocatalysis research.

## Methods

### Synthesis of [Au₂₅(pMBA)₁₈]⁻

[Au₂₅(pMBA)₁₈]⁻ NCs were prepared using CO reduction according to the previous protocol with some minor amendments. In particular, 10 mL of aqueous solution of 50 mM pMBA (in 150 mM NaOH) and 5 mL of 50 mM HAuCl₄ were added into 238.75 mL of ultrapure water sequentially. Then, the mixed solution was stirred for 5 min. After that, the pH of the reaction mixture was brought up to 11.05 by dropping in 1.0 M NaOH aqueous solution and then was stirred for another 10 min to form Au(I)-(pMBA) complex. In sequence, CO was bubbled into the reaction mixture for 2 min to initiate the reduction of Au(I)-(p-MBA) complex. The mixed solution was allowed for proceeding air-tightly for 7 days at room temperature (25 °C) and under vigorous stirring. The reddish-brown solution was obtained at the end of this procedure as raw products.

### Synthesis of [Au₂₅(MHA)₁₈]⁻ and [Au₂₅(HCys)₁₈]⁻

[Au₂₅(MHA)₁₈]⁻ NCs were prepared using NaBH₄ reduction according to the previous protocol with some minor amendments. In particular, 100 mL of aqueous solution of 5 mM MHA and 12.5 mL of 20 mM HAuCl₄ were added into 120 mL of ultrapure water for the formation of Au(I)-MHA complex. Then, 12.5 mL of 1.0 M NaOH solution was introduced into the reaction mixture, followed by the addition of 5 mL of 0.1 M NaBH₄ solutions. The mixed solution was stirred for 24 h and the raw products were collected. [Au₂₅(HCys)₁₈]⁻ NCs were synthesized using the same synthetic conditions except some minor changes in experimental parameters including the stirring time for 6 h.

### Synthesis of Au nanoparticles (NPs)

Au NPs were prepared using NaBH₄ reduction. 12.5 mL of 20 mM HAuCl₄ and 5 mL of 50 mM pMBA were added into 120 mL of ultrapure water followed by the addition of 5 mL of 0.1 M NaBH₄ solutions. The precipitate was washed by water and ethanol for three times.

## Purification

The raw products were first concentrated by 10 times via rotary evaporation (water bath temperature 45 °C, cooling temperature 2 °C, and rotation rate 102 r.p.m.). Ethanol (double the volume of the concentrated NC solution) was then added, followed by a centrifugation at 18,353 g. for 5 min. The resultant precipitate was washed with DMF for 2 times and re-dissolved in ultrapure water to form an aqueous solution of purified $[Au_{25}(SR)_{18}]^-$ ([Au] = 10 mM) for further use. For the powder of Au NCs and Au NPs, their aqueous or turbid solutions can be obtained by freeze-drying (−80 °C under the vacuum of 1 bar).

## Electrochemical experimental procedures

All electrochemical measurements were carried out using a standard three-electrode H type cell on Auto-lab electrochemical workstation at room temperature, with an Ag/AgCl as reference electrode and Pt foil as counter electrode. The half cells were separated by a piece of proton exchange membrane (Nafion 117) and filled with 1.0 M KOH as electrolyte. Before the OER experiment, the electrolyte was saturated with $O_2$ for 15 min. For the preparation of thin-film working electrode, the catalyst ink solutions were loaded on the carbon paper (1 cm²) with a loading amount of 0.026 nmol $Au_{25}$ NCs. The catalyst ink solutions were prepared by dispersing certain amount of catalysts in N-Methyl-2-pyrrolidone (NMP) with Nafion (5 wt%). Taking $[Au_{25}(pMBA)_{18}]^-$ NCs as an example, ~2 mg $[Au_{25}(pMBA)_{18}]^-$ powder and 2 mg of VulcanXC-72 were dispersed in 980 μL of NMP and followed by adding 20 μL of Nafion(5 wt%) and stirred this mixed solution until the formation of homogenous ink. Then, the electrode was dried in vacuum drying oven with 30 mbar and 60 °C overnight for further measurement. Linear sweep voltammetry (LSV) and cyclic voltammetry (CV) analysis was carried out of 5 mV s⁻¹ with iR correction.

## In situ electrochemical-Raman tests

In situ Raman scatting spectroscopy measurements are carried out at room temperature using a micro-Raman spectrometer (Horiba JY-T64000) in a backscattering configuration. A solid-state laser (λ = 532 nm) and an Air ion laser (λ = 488 nm) have been used to excite the samples. The backscattered signal was collected through a 100× objective and dispersed by a 1800-g/mm grating.

## Characterization

Solution and electrolyte pH was monitored by a Mettler Toledo FE 20 pH-meter. Crude product was centrifuged by an Eppendorf Centrifuge 5424. UV–vis spectra were recorded by a Shimadzu UV-1800 spectrometer with the fast scan speed and single scan mode. X-ray photoelectron spectroscopy (XPS) was conducted on a Kratos AXIS Ultra DLD spectrometer. All binding energies were referenced to the C(1 s) hydrocarbon peak at 284.5 eV. For the transmission electron microscopy (TEM), 10 μL aqueous solution of individual Au NCs (0.01 mM on Au atom basis) or Au NPs ( ~ 0.01 mg/mL) was dropped on the copper gird for TEM, followed by air-drying at room temperature. Cluster concentrations were measured and normalized by ICP-OES on a Thermo Scientific iCAP 6000 ICP-OES. ESI-MS analysis was carried out on a Bruker micro TOF-Q system in negative ion mode. Detailed operating conditions of ESI-MS analysis are given as followings: source temperature/120 °C, dry gas flow rate/4 L per min, nebulizer pressure/0.4 bar, and capillary voltage/3.5 kV.

## Computational details

Differential charge density maps: The spin unrestricted DFT calculations in this study are based on the package of DMol3 8.0[51,52]. The generalized gradient approximation (GGA) implemented in the Perdew–Burke–Ernzerhof (PBE) functional was employed as the exchange-correlation functional[53]. The DFT semi-core pseudopotentials[54] was chosen to describe the core electrons and the valence electrons were considered using a double numerical

basis set including p-polarization function DNP (version 4.4) with the orbital cutoff of 4.5 Å. Dispersion corrected DFT (DFT-D) via the TS scheme was used to describe the dispersion interactions among all the atoms in models[55]. The electronic energy was considered self-consistent when the energy change was smaller than 10⁻⁶ eV. The solvation model COSMO[56] was used to simulate aqueous environment, with the dielectric constant set as 78.54 (which is a value for water). A 0.002 Ha smearing value was added to accelerate convergence.

The $Au_{25}$ NCs was put in a 30 Å × 30 Å × 30 Å cubic cell. The calculation was performed with a Gamma Monkhorst-Pack k-point grid. A geometry optimization was performed with the convergence criteria of the energy change smaller than 10⁻⁵ eV, the gradient change smaller than 0.002 Ha/Å or the displacement change smaller than 0.005 Å. Atomic charges were obtained via the Mulliken method.

## Adsorption energy profiles

Spin-polarized density functional theory (DFT) calculations were performed in the Quickstep module[57] of CP2K[58]. The mixed Gaussian and plane wave (GPW) approach[59] using the Perdew–Burke–Ernzerhof (PBE) functional[60] was employed with the Grimme D3-dispersion correction[61]. The Kohn-Sham orbitals were expanded in the molecularly optimized basis set with a double-ζ Gaussian basis set augmented with a set of p-type polarization functions (MOLOPT-SR-DZVP)[62] with core electrons represented by the Goedecker–Teter–Hutter pseudopotentials[63–65]. In all calculations, the plane-wave kinetic energy cutoff was set to 400 Ry. Convergence threshold of the electronic structure relaxation was set to 10⁻⁶ Hartree and the force convergence criterion of the geometry optimizations was set to $4.5 \times 10^{-4}$ Hartree/Bohr.

We took the models of $[Au_{25}(SCH_3)_{15}(SR)_3]^{3-q}$ (SR = pMBA or MHA, q = 0) to understand the effect of the ligands on the OER performance of $Au_{25}$ nanoclusters in alkaline conditions. In those models, three thiolate ligands were treated explicitly in their anionic form while others are capped by methyl groups. The charge state q = 0 of the Au cluster was chosen to be consistent with the experimental observation.

The computational hydrogen electrode was used to obtain the Gibbs free energy of OER for each elementary step in the form of $\Delta G = \Delta E + \Delta E_{ZPE} - T\Delta S$, in which $\Delta E$ is the difference of DFT total energy, $\Delta E_{ZPE}$ the zero-point energy difference calculated from vibrational frequencies, and $\Delta S$ the entropy difference between the products and reactants for each step. The ZPE and entropy values were adopted from literature[66].

## Data availability

All relevant data are available from the corresponding authors on request.

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

## Acknowledgements

This work is financially supported by the National Natural Science Foundation of China (22071174). The authors acknowledge the financial support from the Ministry of Education, Singapore, Academic Research Fund (AcRF) Grant No. R-279-000-580-112 and A-8000054-01-00. L.W. would like to thank the support from A*STAR (Agency for Science, Technology and Research) under its LCERFI program (Award No U2102d2002), and the support by National Research Foundation (NRF) Singapore, under NRF Fellowship (NRF-NRFF13-2021-0007). DFT calculations of adsorption energies were supported by the U.S. Department of Energy, Office of Science, Office of Basic Energy Sciences, Chemical Sciences, Geosciences, and Biosciences Division, Catalysis Science Program.

## Author contributions

J.X. supervised this work. J.X., Q.Y., and Z.L. conceived the idea, designed the experiments, whereas Z.L. carried out the experiments and characterizations. H.T., and Z.H. performed in situ Raman electrochemical test. L.W. co-supervised this work towards electrocatalysis and provided helpful suggestions. Under the supervision of D.-E.J., B.L. calculated adsorption energies using DFT. Z.L. and Q.Y. wrote the manuscript. All authors discussed the results and commented on the manuscript.

## Competing interests

The authors declare no competing interests.
