## [Peer Review File · Nature Communications]

REVIEWER COMMENTS

Reviewer #1 (Remarks to the Author):

This manuscript describes a study of Au₂₅ nanoclusters with various capping ligands for oxygen evolution. I found the work interesting and potentially impactful. However, I have a few concerns/comments:

- Some stronger evidence (even from previous work) that the particles shape is not changing significantly among the three ligands would be helpful for supporting the main point. This may be hinted at at the end of page 4, but I don't see a clear statement that the three particles have the same shape and how this is known. Further, if there is evidence that the structure is not significantly changed under electrocatalytic conditions, such as in situ or post-reaction characterization, that would strengthen the paper significantly.

- Broadly, the paper would be strengthened by DFT calculations of something that is experimentally measured (right now, the DFT is completely complementary to the experiment). For example, DFT calculations of electron binding energies to compare with XPS, and/or adsorption energies of OH/OOH to compare with the reactivity studies, would give much more support to the microscopic picture. A full set of calculations of the reaction on all three particles would likely be challenging and is probably beyond the scope of the paper, but a few simple calculations for comparison would be good.

- There has been a fair amount of work on the effect of thiolate ligands on catalysis for larger particles. Can any comparison to this work be made?

- It is unclear how the charge density difference maps were created. What two states were used in the difference? Presumably only the difference for certain atoms is shown. On page 7, the authors state that more details are in the supporting information, but I could not find any more details.

- The charge density maps show the charge distribution on Au but not obvious there's a deficiency, as the authors state.

- I would suggest the authors plot turnover frequency vs. one or more properties of the nanoclusters (e.g., calculated Bader charge or XPS shift), if they can find any reasonable correlations. This would probably be best in the supporting information as it is hard to be conclusive with 3 data points, but it could point towards a possible design principle.

- I did not quite follow how the authors ensured the same amount of material is present in the different samples, and how they calculated the TOF. They state they can count the number of surface atoms because they know the structure, but how do they know how much Au is present overall?

- For the caption of Fig 3e, I suggest the authors specify the system studied.

- For Fig 3f, it seems to me that the various potentials should be indicated on the plot.

- A clearer statement with citations that the postulated mechanism is likely to be the correct one would be helpful.

I believe these are all typos that I happened to notice:

- "The above-mentioned different Tafel slop values suggest the RDS is changed from the adsorption of OH⁻ to Au-O-H formation" should this be Au-O-OH?
- "Tafel slop"
- "the adsorption on OH⁻ no longer inhabits"
- "it is widely accepted that the absorbed"
- "The participate was washed"

Reviewer #2 (Remarks to the Author):

The manuscript by Liu et al. discusses a potential effect of electron-withdrawing capping ligands at atomic Au clusters on the rate-determining step (res) of water oxidation. The authors report experimental evidence that the choice of stabilising ligand on the nano cluster influences the rds of water oxidation in alkaline media.

- Will the work be of significance to the field and related fields? How does it compare to the established literature? If the work is not original, please provide relevant references.

In general, I think ligand engineering is a maybe obvious, yet very interesting route to achieve improved electrocatalytic behaviour of nanocatalysts. Currently, however, the work still needs to be enhanced further to answer pressing questions about the system under study as outlined below.

- Does the work support the conclusions and claims, or is additional evidence needed?

Open questions:

The most pressing question I have regards the reaction mechanism. The authors claim that sulfur-bound ligands stabilise the Au cluster and at the same time, depending on the ligands' electron withdrawing abilities, change the Au(I) positive charge, in this way affecting the electrocatalytic ability of the Au(I), i.e.-OH adsorption site. I wonder how strong the chemisorption(?), or interaction, of the ligands with the Au cluster is and how a change in adsorption strength will affect the rds of water oxidation. I assume

that the ligands have to be replaced by OH during the reaction, at least partially, to free Au(I) active sites? Unfortunately, in Figure 5, the mechanism schematic, the ligands have been 'omitted... for clarity', which seems rather unfortunate as they are said to play the crucial activating role. The authors should present evidence for the fate / chemical changes of the ligands during the reaction.

This could, for example, be accomplished by investigating the in situ Raman data. Unfortunately, Figure 4 only shows a rather small extraction ranging from 550 to 850 cm^{-1} . The region around 200-300 cm^{-1} would provide insights into the Au-S or Au-ligand bond (and its fate during water oxidation or -OH adsorption). Similarly, the higher energy range 800-2000 cm^{-1} should exhibit modes related to the aromatic or aliphatic CC modes of the ligands, which give rich information about the surface chemistry at the Au nano clusters. These spectral regions should be investigated in detail to understand the ongoing chemistry. Also, the inspected region between 550-850 cm^{-1} gives extremely broad responses. Why is that? What effect of the ligand-dependent differing positive Au(I) charge do the authors expect (and observe) in the Raman response?

The effect of OH/ligand exchange on the electrochemical data should also be discussed.

A question regarding Figure 3 / text page 8: Where do the over potential values (line 193, the 540 mV supposedly belong to the MHA-ligand cluster? typo twice HCys?) stem from? How were they extracted from the Figure (3a? or which?)? In Figure 3a, the MHA-10 mA/cm^2 is hardly in the figure anymore, it seems. A discussion on how these overpotential values compare to literature should be included for the reader to understand the quantitative (potential) impact of ligand tuning on the reactivity. What are the pure (ligand-free) Au overpotential values?

What are the error bars of the generated O₂ volume data points in Figure 3e?

What are the applied potentials in Figure 3f?

Regarding the Tafel slopes, the discussion on page 9 should be more detailed. In line 222, the authors state that the slopes for MHA and HCys ligands are 'a little higher' than for pMBA ligands. In fact, they are a factor 3 and 5 higher, respectively, which is quite a lot. What do these values mean? How do the authors deduce in such a straightforward way the intermediates that form during the reaction, given the strong electronic ligand effect? What does the sentence 'additional overpotentials ... were required to stabilise Au-OH for adequate coverage'?

In the discussion of the mechanism, p13, line 315ff, the authors discuss a 'threshold [OH] coverage'. What does this mean? Why is it needed? Why not correlate this statement with the Raman data? How

do the authors distinguish Au-O, Au-OH and Au-OOH species? How is the ligand-O(H) place exchange entangled with the coverage discussion?

Last paragraph should probably be called 'summary' rather than discussion.

- Are there any flaws in the data analysis, interpretation and conclusions? Do these prohibit publication or require revision?
- Is the methodology sound? Does the work meet the expected standards in your field?

More detailed discussion and analysis is necessary in my opinion.

- Is there enough detail provided in the methods for the work to be reproduced?

Yes, I should think so.

Where applicable, reporting summaries are requested from the authors to improve the transparency and reproducibility of published results. We hope the file, if included, will aid in your evaluation of the paper as they contain key information pertaining to study design and analysis.

I would be happy to review a heavily improved manuscript at a later stage.

Reviewer #3 (Remarks to the Author):

The authors reported the ligand effect of atomically precise gold nanoclusters (NCs) on electrocatalytic OER kinetics. Three well-defined model Au₂₅ NCs capped by MHA, HCys, and pMBA were studied. The Au₂₅(pMBA)₁₈ exhibited the highest OER activity among the three NCs, which was ascribed to more partial positive charge on Au(I) induced by pMBA with a stronger electron-withdrawing ability. Different RDSs were suggested based on the Tafel analysis and in-situ Raman spectroscopy. This is an interesting result showing the ligand effect on the electrocatalysis RDS, but some evidence needs to be strengthened to support the conclusions. The manuscript is recommended for consideration after addressing the following concerns.

1) The authors suggested different RDS of OER for Au₂₅ NCs capped by MHA, HCys, and pMBA based on the in-situ Raman spectra. What is the rationale to connect the reaction intermediate with the reaction RDS? Observation of reaction intermediate does not necessary indicate the reaction RDS.

2) The thiolate-protected Au NCs may undergo dethiolation process during electrocatalytic reaction. Is the integrity of Au₂₅(pMBA)₁₈ preserved after electrocatalytic reaction?

3) The charge state of Au₂₅ NCs should be carefully determined in comparison with their open-circuit potential corresponding to the anionic form, i.e., Au₂₅(-1). Please indicate the open-circuit potentials on the cyclic voltammograms and determine the charge state with respect to the open-circuit potential.

4) On Page 12, line 277: “the quantum size effect” is quite ambiguous. How does the quantum size effect affect the Raman shift?

Replies to reviewers' comments and descriptions of revisions made

Comments by Reviewer #1:

This manuscript describes a study of Au₂₅ nanoclusters with various capping ligands for oxygen evolution. I found the work interesting and potentially impactful. However, I have a few concerns/comments:

Reply: We are excited about the reviewer's positive appreciation of the novelty and significance of our work. Indeed, we believe the correlations between the molecular-level structure and performance can provide an in-depth understanding on the origin of the electrocatalytic activity and selectivity of metal nanoclusters. This is the intrinsic motivation of the present work. We also believe that the ligand engineering-based optimizations for clusters' catalytic performance will gain broad fundamental and practical interests from multidisciplinary audiences of *Nature Communications*. This will further trigger more research activities in the diverse fields of cluster chemistry, nanochemistry and electrochemistry. We thank the reviewer for his/her inspiring and instructive comments/suggestions, which have been taken into careful consideration in this revision. Please see following the point-to-point response to the reviewer's comments/suggestions.

1. Some stronger evidence (even from previous work) that the particles shape is not changing significantly among the three ligands would be helpful for supporting the main point. This may be hinted at the end of page 4, but I don't see a clear statement that the three particles have the same shape and how this is known. Further, if there is evidence that the structure is not significantly changed under electrocatalytic conditions, such as in situ or post-reaction characterization, that would strengthen the paper significantly.

Reply: Thanks for your kind suggestion. We agree with the reviewer that the particle shape is a key factor dictating the electrocatalytic performance of metal nanoparticles/nanoclusters (NCs). Due to the strong Au-S bonds, Au₂₅(SR)₁₈ NCs (SR = thiolate ligands) have been widely documented with similar Au-S framework regardless of the R tails (*Nanoscale* **2018**, 10, 10758–10834; *ACS Nano* **2016**, 10, 7998–8005). The similar morphology (*i.e.*, Au-S framework) of Au NCs capped by

different thiolate ligands in this work is also evidenced by their similar UV-vis absorption spectra (Fig. 1, extracted as Fig. RL-1 below for easy identification), which are commonly regarded as optical fingerprint of metal NCs due to their atomic-level sensitivity to Au-S structure (*J. Am. Chem. Soc.* **2008**, 130, 5883–5885).

Fig. RL-1 (also as Fig. 1) (a) Schematic illustration and (b) ultraviolet-visible absorption spectra of Au₂₅ NCs capped by different ligands.

Based on the reviewer's suggestion, we conducted additional experiment to confirm the structural stability of Au₂₅ NCs during the electrocatalytic reactions. We performed UV-vis absorption spectroscopy and ESI-MS to determine the formula of Au NCs before and after oxygen evolution reaction (OER). The UV-vis absorption spectra before and after 200 cycles of LSV tests show almost identical profiles with minor differences in the intensity, where slight decrease in absorption intensity was observed at ~695 nm (Fig. RL-2a also as Supplementary Fig. 21). This reflects that the majority of [Au₂₅(pMBA)₁₈]⁻ survive after the OER although a trivial degradation is not avoidable. The ESI-MS of Au₂₅ NCs capped by pMBA after OER process further confirm their unchanged formulae after OER (Fig. RL-2b also as Supplementary Fig. 22). With respect to [Au₂₅(MHA)₁₈]⁻ and [Au₂₅(HCys)₁₈]⁻ NCs, more distinct decrease in absorption peak intensity was observed. (Fig. RL-2c and 2e also as Supplementary Fig. 23 and 24), indicating that they are less stable than [Au₂₅(pMBA)₁₈]⁻. Nevertheless, their formulae are [Au₂₅(MHA)₁₈]⁰ and [Au₂₅(HCys)₁₈]⁰ NCs, which are unearthed by ESI-MS (Fig. RL-2d and 2f also as Supplementary Fig. 25 and 26). Overall, the atomic-level morphology is not changed for Au₂₅ NCs capped by these three ligands before and after the OER test.

Fig. RL-2 (a, c, e) UV-vis absorption spectra; (b, d, f) ESI-MS of $[\text{Au}_{25}(\text{SR})_{18}]^-$ before and after OER. The filled curves in the insets are simulated isotope patterns of labelled cluster formulae.

Revisions:

Supplementary Information (SI), Page 24-29:

Fig. RL-2 is included in SI as Supplementary Fig. 21-26.

Page 11, Lines 3-19:

‘In addition, we conducted additional experiment to confirm the structural stability of Au_{25} NCs during the electrocatalytic reactions. We performed UV-vis absorption spectroscopy and ESI-MS to determine the formula of Au NCs before and after oxygen evolution reaction (OER). The UV-vis absorption spectra before and after 200 cycles of LSV tests show almost identical profiles with minor differences in the intensity, where slight decrease in absorption intensity was observed at ~ 695 nm (Supplementary Fig. 21). This reflects that the majority of $[\text{Au}_{25}(\text{pMBA})_{18}]^-$ survive after the OER although a trivial degradation is not avoidable. The ESI-MS of Au_{25} NCs capped by pMBA after OER process further confirm their unchanged formulae after OER (Supplementary Fig. 22). With respect to $[\text{Au}_{25}(\text{MHA})_{18}]^-$ and $[\text{Au}_{25}(\text{HCys})_{18}]^-$ NCs, more distinct decrease in absorption peak intensity was observed. (Supplementary Fig. 23 and 24), indicating that they are less stable than

$[\text{Au}_{25}(\text{pMBA})_{18}]^-$. Nevertheless, their formulae are $[\text{Au}_{25}(\text{MHA})_{18}]^0$ and $[\text{Au}_{25}(\text{HCys})_{18}]^0$ NCs, which are unearthed by ESI-MS (Supplementary Fig. 25 and 26). Overall, the atomic-level morphology is not changed for Au_{25} NCs capped by these three ligands before and after the OER test.’

2. Broadly, the paper would be strengthened by DFT calculations of something that is experimentally measured (right now, the DFT is completely complementary to the experiment). For example, DFT calculations of electron binding energies to compare with XPS, and/or adsorption energies of OH/OOH to compare with the reactivity studies, would give much more support to the microscopic picture. A full set of calculations of the reaction on all three particles would likely be challenging and is probably beyond the scope of the paper, but a few simple calculations for comparison would be good.

Reply: Thanks for your advice. We have performed additional DFT calculation accordingly. The discussion on the adsorption energy profiles have been included in the revised manuscript.

Fig. RL-3 (also as Supplementary Fig. 28 and 29) Calculated adsorption energy profiles of OH and O on Au₂₅ NCs capped by pMBA and MHA.

Revisions:

Supplementary Information (SI), Page 31 and 32:

The Fig. RL-3 has been included in the SI as Supplementary Fig. 28 and 29.

Page 14, Lines 13-19 and Page 15, Lines 1-4:

‘To understand the ligand effect on the OER performance of Au₂₅ NCs in alkaline conditions, the adsorption energies of OH and O on Au₂₅ NCs were calculated to compare pMBA and MHA ligands. To save the computational cost, we employed the models of [Au₂₅(SCH₃)₁₅(SR)₃]³⁻ (SR = pMBA or MHA) where three deprotonated real thiolate ligands are present on the same dimeric motif while other thiolate ligands are simplified as SCH₃ (more computational details can be found in the section Methods). As shown in supplementary Fig. 28 and 29, both O and OH adsorb on an Au(I) atom in the dimeric motif and, more importantly, their adsorption is less stronger with the presence of the pMBA ligand than the MHA ligand, indicating that *OH and *O species are easier to be activated on [Au₂₅(pMBA)₁₈]⁻. Such correlations of adsorption energy with OER activities have been observed by others as well^{48,49}.’ (*Nat Catal.* **2018**, 1, 300–305; *ACS Catal.* **2022**, 12, 11597–11605)

3. There has been a fair amount of work on the effect of thiolate ligands on catalysis for larger particles. Can any comparison to this work be made?

Reply: Thanks for your kind suggestion. To address this comment, we also compared the electrocatalytic performance of [Au₂₅(MHA)₁₈]⁻ with that of the larger Au nanoparticles capped by pMBA (NPs, ~5 nm in diameter, prepared by NaBH₄ reduction (Supplementary Fig. 8). The LSV curves in Supplementary Fig. 9 (extracted as Fig. RL-4 below) indicated that the highest current density achieved by Au NPs within the same potential window from 1.1 to 1.8 V is only up to 2.5 mA/cm². In contrast, the

$[\text{Au}_{25}(\text{MHA})_{18}]^-$ NC, which performs the worst among the synthesized Au_{25} NCs delivers $\sim 11 \text{ mA/cm}^2$ when the potential increases to 1.8 V. This comparison reflects that the electrocatalytic performance of Au_{25} NCs are superior to the large counterparts. Moreover, we also compared our Au_{25} NCs with thiolate-protected transition metal electrocatalysts reported elsewhere in terms of the overpotential required for reaching a current density of 10 mA/cm^2 (Fig. RL-5 also as Supplementary Fig. 10). As can be seen, $[\text{Au}_{25}(\text{pMBA})_{18}]^-$ NCs display a better performance, while $[\text{Au}_{25}(\text{HCys})_{18}]^-$ and $[\text{Au}_{25}(\text{MHA})_{18}]^-$ are comparable to other transition metal based electrocatalysts.

Fig. RL-4 (also as Supplementary Fig. 9) LSV curve of comparison for $[\text{Au}_{25}(\text{SR})_{18}]^-$ NCs and Au NPs capped by pMBA.

Fig. RL-5 (also as Supplementary Fig. 10) Comparison of overpotentials required for reaching a current density of 10 mA/cm². (Ni₆(2-phenylethanethiol)₁₂, Ni₅(2-phenylethanethiol)₁₀, and Ni₄(2-phenylethanethiol)₈ (*Inorg. Chem.* **2023**, 62, 1875–1884); Pd₆(SC₁₂H₂₆S)₁₂ (*Chem. Commun.* **2017**, 53, 9733–9736); Au₂₅(PET)₁₈ (*Nanoscale*, **2020**, 12, 9969–9979); Copper (I)-thiolate cluster (*Chem. Commun.* **2020**, 56, 3967–3970), Ni₆(phenylethyl)₁₂ (*ACS Catal.*, **2016**, 6, 1225–1234); The overpotentials in this work are marked in stars filled with different colors.

Revisions:

Supplementary Information (SI), Page 13, Supplementary Fig. 10:

The RL-5 is included in SI as Supplementary Fig. 10.

Page 9, Lines 9-14:

‘Moreover, we also compared our Au₂₅ NCs with other thiolate-protected transition metal electrocatalysts in terms of the overpotential required for reaching a current density of 10 mA/cm² (Supplementary Fig. 10). As can be seen, [Au₂₅(pMBA)₁₈]⁻ NCs display a better performance, while [Au₂₅(HCys)₁₈]⁻ and [Au₂₅(MHA)₁₈]⁻ are comparable to other transition metal based electrocatalysts.’

4. It is unclear how the charge density difference maps were created. What two states were used in the difference? Presumably only the difference for certain atoms is shown. On page 7, the authors state that more details are in the supporting information, but I could not find any more details.

Reply: Sorry for the misleading. We have added the calculation details in the section Methods in the revised manuscript. As it is too computationally demanding to map out all 18 thiolate ligands on the surface of Au₂₅ NCs, one representative ligand was modelled for individual Au₂₅ in our DFT calculations. The spin unrestricted DFT calculations in this study are based on the package of DMol3 8.0. The generalized gradient approximation (GGA) implemented in the Perdew-Burke-Ernzerhof (PBE) functional was employed as the exchange-correlation functionality. The DFT semi-core pseudopotentials was chosen to describe the core electrons and the valence electrons were considered using a double numerical basis set including p-polarization function DNP (version 4.4) with the orbital cutoff of 4.5 Å. Dispersion corrected DFT (DFT-D)

via the TS scheme was used to describe the dispersion interactions among all the atoms in models. The electronic energy was considered self-consistent when the energy change was smaller than 10^{-6} eV. The solvation model COSMO was used to simulate aqueous environment, with a dielectric constant set of 78 (which is the value for water). A 0.002 Ha smearing value was added to accelerate convergence.

The Au₂₅ NCs were placed in a 30 Å×30 Å×30 Å cubic cell. The calculation was performed with a Gamma Monkhorst-Pack k-point grid. A geometry optimization was performed with the convergence criteria of the energy change smaller than 10^{-5} eV, the gradient change smaller than 0.002 Ha/Å or the displacement change smaller than 0.005 Å. Atomic charges were obtained via the Mulliken method.

5. The charge density maps show the charge distribution on Au but not obvious there's a deficiency, as the authors state.

Reply: Thank you for spotting this issue out. To view the charge distribution in a clearer manner, we have zoomed in the charge density maps. Zoom-in views for the charge density maps of Au(I) in the Au₂₅ NC models are now included in the revised Fig. 2. As it is too computationally demanding to map out all 18 thiolate ligands on the surface of Au₂₅ NCs, one representative ligand was modelled for individual Au₂₅ in our DFT calculations. The differential charge density maps in Fig. RL-6 (also as Fig. 2) suggest a charge accumulation/deficiency for sulfur and the correspondingly bonded gold (as shown in the dotted red circles), respectively. The Au(I) exhibits a more obvious charge deficiency in [Au₂₅(pMBA)₁₈]⁻ NCs, which should be attributed to stronger electron-withdrawing capability of pMBA. In stark contrast, the charge depletion of Au(I) is relatively weaker for [Au₂₅(HCys)₁₈]⁻ and [Au₂₅(MHA)₁₈]⁻.

Revisions:

Page 7, Fig. 2:

Fig. RL-6 (also as Fig. 2) Differential charge density maps of Au_{25} NCs capped by different ligands. Only one representative thiolate ligand is modelled in individual cluster for reasonable computational cost. A charge accumulation/deficiency for sulfur and the correspondingly bonded gold is highlighted. The insets are zoom-in views on Au(I) showing its charge deficiency.

6. I would suggest the authors plot turnover frequency vs. one or more properties of the nanoclusters (e.g., calculated Bader charge or XPS shift), if they can find any reasonable correlations. This would probably be best in the supporting information as it is hard to be conclusive with 3 data points, but it could point towards a possible design principle.

Reply: Thanks a lot for the discerning suggestions. We have plotted the turn-over frequency (TOF) of Au_{25} NCs against their Au(I)/Au(0) ratio obtained from deconvolution of XPS peaks. As can be seen in Fig. RL-7, an exponential ascending trend is observed for TOF with increasing Au(I)/Au(0) ratio. Fig. RL-7 is now included as Supplementary Fig. 18.

Fig. RL-7 (also as Supplementary Fig. 18) (a) Au(I)/Au(0) ratio for [Au₂₅(SR)₁₈]⁻ NCs. (b) TOF vs. Au (I)/Au(0) ratio for [Au₂₅(SR)₁₈]⁻ NCs.

Revisions:

Supplementary Information (SI), Page 21, Supplementary Fig. 18:

Fig. RL-7 is included in SI as Supplementary Fig. 18.

Page 10, Lines 18-24:

‘In order to correlate the Au charge deficiency of Au₂₅ NCs to their turn-over frequency (TOF) in OER, we determined the Au(I)/Au(0) ratio in individual Au₂₅ NCs by deconvoluting Au 4f peak in their XPS spectra. Then we plotted TOF vs. Au(I)/Au(0) ratio in Supplementary Fig. 18. As can be seen in Supplementary Fig. 18, an ascending trend is observed for TOF with increasing Au(I)/Au(0) ratio, in good consistency with electron-withdrawing capability of examined thiolate ligands.’

7. I did not quite follow how the authors ensured the same amount of material is present in the different samples, and how they calculated the TOF. They state they can count the number of surface atoms because they know the structure, but how do they know how much Au is present overall?

Reply: We are sorry for the ambiguity in our last submission. The TOF is calculated based on the equation, $TOF = I/(4F*N)$, where I, F, and N are the current, Faraday constant and the molar number of active sites, respectively. The moles of the active sites are estimated by assuming all 12 Au atoms in the surface motifs of individual [Au₂₅(SR)₁₈]⁻ are equivalent active sites. The concentration of [Au₂₅(SR)₁₈]⁻ was

measured by inductively coupled plasma optical emission spectroscopy (ICP-OES) on a Thermo Scientific iCAP 6000. The loading amount of $[\text{Au}_{25}(\text{SR})_{18}]^-$ in each sample is kept consistent in terms of their molar number. The calculation details have been included in the SI accordingly.

Revisions:

Supplementary Information (SI), Page 2, Lines 17-25:

‘The turnover frequency(s^{-1}) can be estimated from:

$$\text{TOF} = I / (4F * N)$$

where I is the current density (A cm^{-2}), can be calculated with 1 cm^2 working area for the different electrodes during the LSV measurement in 1.0 M KOH, F is the Faraday constant (C mol^{-1}), and N is the molar number of active sites for the different electrodes. The molar number of active sites is estimated by assuming all 12 Au atoms in the surface motifs of individual $[\text{Au}_{25}(\text{SR})_{18}]^-$ are equivalent active sites. The concentration of $[\text{Au}_{25}(\text{SR})_{18}]^-$ was measured by inductively coupled plasma optical emission spectroscopy (ICP-OES) on a Thermo Scientific iCAP 6000.’

8. *For the caption of Fig 3e, I suggest the authors specify the system studied.*

Reply: Thanks for your kind comments. We have revised the caption of Fig. 3e accordingly.

Revisions:

Page 8, caption of Fig. 3e:

‘Fig. 3e O_2 production volumes as a function of water-splitting time by using $[\text{Au}_{25}(\text{pMBA})_{18}]^-$ as electrocatalysts: the circles are experimentally obtained O_2 volumes, while the solid line indicates theoretical value of O_2 calculated by assuming a 100% Faradaic efficiency for the anode reaction at the current density of 10 mA/cm^2 .’

9. *For Fig 3f, it seems to me that the various potentials should be indicated on the plot.*

Reply: We appreciate the reviewer's kind suggestion. We have marked all the applied potentials (i.e., 1.57, 1.62, 1.66 and 1.59 V vs. RHE) in Fig. RL-8 (also as Fig. 3f in the revised manuscript).

Fig. RL-8 (also as Fig. 3f) I-t curves of $[\text{Au}_{25}(\text{pMBA})_{18}]^{-}$ NCs with different applied electrode potentials. The electrocatalytic reactions were carried out in O_2 -saturated aqueous solution of 1.0 M KOH.

Revisions:

Page 8, Fig. 3f:

Fig. RL-8 is included in the main text as Fig. 3f.

10. A clearer statement with citations that the postulated mechanism is likely to be the correct one would be helpful.

Reply: This is another very constructive suggestion for improving the readability of our manuscript. Accordingly, we have added some references which is supportive to our mechanism discussion.

Revisions:

Page 22, Lines 26-28 and Page 23, Lines 10-20:

39. Sun S, *et al.* Switch of the Rate-Determining Step of Water Oxidation by Spin-Selected Electron Transfer in Spinel Oxides. *Chem. Mater.* **31**, 8106–8111 (2019).

46. Shinagawa T, Garcia-Esparza AT, Takanebe K. Insight on Tafel slopes from a Microkinetic Analysis of Aqueous Electrocatalysis for Energy Conversion. *Sci. Rep.* **5**, 13801 (2015).

47. Yang S, Hetterscheid DGH. Redefinition of the Active Species and the Mechanism of the Oxygen Evolution Reaction on Gold Oxide. *ACS Catal.* **10**, 12582–12589 (2020).
48. Li T, et al. Atomic-scale Insights into Surface Species of Electrocatalysts in Three Dimensions. *Nat. Catal.* **1**, 300–305 (2018).
49. Chandra D, et al. Highly Efficient Electrocatalysis and Mechanistic Investigation of Intermediate $\text{IrO}_x(\text{OH})_y$ Nanoparticle Films for Water Oxidation. *ACS Catal.* **6**, 3946–3954 (2016).

11. I believe these are all typos that I happened to notice:

- *“The above-mentioned different Tafel slop values suggest the RDS is changed from the adsorption of OH^- to Au–O–H formation” should this be Au–O–OH?*
- *“Tafel slop”*
- *“the adsorption on OH^- no longer inhabits”*
- *“it is widely accepted that the absorbed*
- *“The participate was washed”*

Reply: We appreciate the reviewer’s careful review. We have corrected all the typos in the revised manuscript.

Comments by Reviewer #2:

The manuscript by Liu et al. discusses a potential effect of electron-withdrawing capping ligands at atomic Au clusters on the rate-determining step (res) of water oxidation. The authors report experimental evidence that the choice of stabilising ligand on the nano cluster influences the RDS of water oxidation in alkaline media. In general, I think ligand engineering is a maybe obvious, yet very interesting route to achieve improved electrocatalytic behaviour of nanocatalysts. Currently, however, the work still needs to be enhanced further to answer pressing questions about the system under study as outlined below.

Reply: We are glad that the reviewer finds our work important and interesting. Indeed, the correlations between molecular- or atomic-level structure and electrocatalytic performance are vital for designing metal NCs for effective electrocatalysis. The atomically precise structure and composition of metal NCs can provide a good platform to clearly illustrate such ‘structure-performance’ relationship for metal nanocatalyst. Moreover, unearthing the in-depth principles (e.g., how to tailor the atomic- or molecular-level structure to tune the kinetics) for electrocatalytic performance enhancement is equally important for adding to metal NCs’ acceptance in many practical catalytic scenarios. The molecular-level surface modification strategy established in this work will gain broad fundamental and practical interest from heterogenous readers of *Nature Communications*, triggering more research activities in diverse fields of cluster chemistry, nano chemistry, and electrochemistry. We also thank the reviewer for his/her inspiring and instructive suggestions. We have taken them into careful consideration in this revision. Please see below for a point-to-point response to the reviewer’s specific comments/suggestions.

1. The most pressing question I have regards the reaction mechanism. The authors claim that sulfur-bound ligands stabilise the Au cluster and at the same time, depending on the ligands’ electron withdrawing abilities, change the Au(I) positive charge, in this way affecting the electrocatalytic ability of the Au(I), i.e.-OH adsorption site. I wonder how strong the chemisorption(?), or interaction, of the ligands with the Au cluster is and how a change in adsorption strength will affect the rds of water oxidation. I assume that the ligands have to be replaced by OH during the reaction, at least partially, to

free Au(I) active sites? Unfortunately, in Figure 5, the mechanism schematic, the ligands have been 'omitted... for clarity', which seems rather unfortunate as they are said to play the crucial activating role. The authors should present evidence for the fate / chemical changes of the ligands during the reaction. This could, for example, be accomplished by investigating the in situ Raman data. Unfortunately, Figure 4 only shows a rather small extraction ranging from 550 to 850 cm^{-1} . The region around 200-300 cm^{-1} would provide insights into the Au-S or Au-ligand bond (and its fate during water oxidation or -OH adsorption). Similarly, the higher energy range 800-2000 cm^{-1} should exhibit modes related to the aromatic or aliphatic CC modes of the ligands, which give rich information about the surface chemistry at the Au nano clusters. These spectral regions should be investigated in detail to understand the ongoing chemistry. Also, the inspected region between 550-850 cm^{-1} gives extremely broad responses. Why is that? What effect of the ligand-dependent differing positive Au(I) charge do the authors expect (and observe) in the Raman response?

Reply: Thanks for your insightful comments. To unambiguously reveal the fate of thiolate ligands in the OER process, we performed UV-vis absorption spectroscopy and ESI-MS on Au₂₅ NCs before and after the electrocatalytic reactions. The UV-vis spectra of [Au₂₅(pMBA)₁₈]⁻ NCs before and after 200 cycles of LSV tests show almost identical profiles with minor differences in the intensity, where slight decrease in absorption intensity was observed at ~695 nm (Fig. RL-9a also as Supplementary Fig. 21). Given UV-vis absorption spectra can be optical fingerprints of Au NCs, the unchanged UV-vis absorption profiles suggest that the majority of [Au₂₅(pMBA)₁₈]⁻ survive after the OER although a trivial degradation is not avoidable. The ESI-MS of Au₂₅ NCs capped by pMBA after OER process further confirm their unchanged formulae after OER (Fig. RL-9b also as Supplementary Fig. 22). With respect to [Au₂₅(MHA)₁₈]⁻ and [Au₂₅(HCys)₁₈]⁻ NCs, more distinct decrease in absorption peak intensity was observed. (Fig. RL-9c and 9e also as Supplementary Fig. 23 and 24), indicating that they are less stable than [Au₂₅(pMBA)₁₈]⁻. Nevertheless, their formulae are [Au₂₅(MHA)₁₈]⁰ and [Au₂₅(HCys)₁₈]⁰ NCs, which are unearthed by ESI-MS (Fig. RL-9d and 9f also as Supplementary Fig. 25 and 26). The combined UV-vis absorption and ESI-MS analyses confirm that the thiolate ligand did not desorb from the surface of Au₂₅ NCs. The extraordinary stability of thiolate ligands on the surface of Au NCs

should be attributed to the strong Au-S bonds, whose dissociation energy could be as high as 300 kJ/mol (*J. Phys. Chem. A* **2015**, 119, 11659–11667). Moreover, the surface Au(I) is largely under-coordinated in gold NCs. Therefore, it is not necessary to remove thiolate ligand to expose Au(I) as active site. Extensive examples have been documented that Au NCs can exhibit remarkable catalytic activities without removing their surface ligands (*Chem. Rev.* **2021**, 121, 567–648).

Fig. RL-9 (a, c, e) UV-vis absorption spectra. (b, d, f) ESI-MS of $[\text{Au}_{25}(\text{SR})_{18}]^{-}$ before and after homogenous OER. The filled curves in the insets are simulated isotope patterns of labelled cluster formulae.

As for Fig. 5 (extracted below as Fig. RL-10 for easy identification), we have added the structure scheme with ligand to make it clearer according to the reviewer's comments.

Fig. RL-10 (also as Fig. 5) Proposed schematic illustration of OER pathways of $[\text{Au}_{25}(\text{SR})_{18}]^-$ NCs. $[\text{Au}_{25}(\text{SR})_{18}]^-$ have the same OER pathways but different RDS caused by varied ligands.

Following the instructive suggestions from the reviewer, we conducted additional *in situ* Raman experiment to monitor the Au-S signal in the same applied potential window as that in Fig. 4a, 4b, and 4c. As shown in Fig. RL-11, similar peaks situated at $\sim 263 \text{ cm}^{-1}$ were observed through the entire potential window, which should be attributed to the Au-S bond in Au₂₅ NCs (*Acc. Chem. Res.* **2018**, 51, 2811–2819). The intensity for the A-S bonds stays almost unchanged, suggesting the ligand maintained during the OER process. Unfortunately, the Raman signal for the ligand body (the aromatic or aliphatic C-C modes) is severely overlapped with the signal of the carbon paper substrate, which makes it hard to interpret the ligand structure evolution during the OER process.

Fig. RL-11 *In situ* Raman signal intensity recorded at varied applied potentials for (a) $[\text{Au}_{25}(\text{MHA})_{18}]^{-}$, (b) $[\text{Au}_{25}(\text{HCys})_{18}]^{-}$, and (c) $[\text{Au}_{25}(\text{pMBA})_{18}]^{-}$ NCs.

In addition, the adsorption energy of OH for Au_{25} NCs was also calculated. We took the models of $[\text{Au}_{25}(\text{SCH}_3)_{15}(\text{SR})_3]^{3-}$ ($\text{SR} = \text{pMBA}$ or MHA) to understand the effect of the ligands on the OER performance of Au_{25} nanoclusters in alkaline conditions. To save the computational cost, we employed the models of $[\text{Au}_{25}(\text{SCH}_3)_{15}(\text{SR})_3]^{3-}$ ($\text{SR} = \text{pMBA}$ or MHA) where three deprotonated real thiolate ligands are present on the same dimeric motif while other thiolate ligands are simplified as SCH_3 . The sample $[\text{Au}_{25}(\text{pMBA})_{18}]^{-}$ and $[\text{Au}_{25}(\text{MHA})_{18}]^{-}$ were selected for exhibiting the best and worst electrocatalytic performance, respectively (more computational details can be found in the section Methods). Fig. RL-12 (also as Supplementary Fig. 28) shows that OH adsorb on an Au(I) atom in the dimeric motif and, more importantly, their adsorption is less stronger with the presence of the pMBA ligand than the MHA ligand, indicating that $^*\text{OH}$ species are easier to be activated on $[\text{Au}_{25}(\text{pMBA})_{18}]^{-}$.

Fig. RL-12 (also as Supplementary Fig. 28). Calculated adsorption energy profiles of OH on Au_{25} NCs capped by pMBA and MHA.

Regarding the broad Raman signal response, we also noticed similar peak broadening in previous reports (*J. Raman Spectrosc.* **1999**, 30, 413–415; *ACS Catal.* **2020**, 10, 12582–12589; *ChemPhysChem* **2010**, 11, 1854–1857). Although the unambiguous cause for such peak broadening is not known to the community, a couple of possible causes are provided as follows. First, the broad Raman bands are probably caused by different hydration states (*J. Electroanal. Chem.* **1986**, 209, 377–386). Second, Raman signal is easily influenced by the adsorbed electrolyte and/or the applied potentials (*ChemPhysChem* **2010**, 11, 1854–1857). Also, the broad response can be attributed to the size effect of Au NCs compared with their bulk counterparts (*J. Raman Spectrosc.* **1999**, 30, 413–415). In bulk materials or large counterparts, the vibrational modes are well defined and narrow due to the free propagating space (*J. Raman Spectrosc.* **2007**, 38, 618–633). However, due to the high surface-to-volume ratio in nanocluster, the phonons are confined to a small volume, where the particle size is comparable to the wavelengths of phonons. As a result, the vibrational modes become broader and more complex.

In summary, we employed *in situ* Raman spectroscopy to monitor the key intermediates on the surface of Au₂₅ NCs in the OER process, which can shed fundamental light on the RDS of OER in the presence of different-thiolate-protected Au₂₅ NCs. Due to the varied electron-withdrawing ability of thiolate ligands, the RDS associated with corresponding Au₂₅ NCs was altered. Specially, for [Au₂₅(MHA)₁₈]⁻ and [Au₂₅(HCys)₁₈]⁻ NCs, Au-OH vibration was detected, while Au-O-OH signal was recorded on [Au₂₅(pMBA)₁₈]⁻ NCs, evidencing the varied RDS on the latter. Thanks to the reviewer's good suggestion, we also managed to evidence the maintained Au-S structure in [Au₂₅(SR)₁₈]⁻ throughout the OER process by *in situ* Raman spectroscopy.

Revisions:

Supplementary Information (SI), Page 24-29:

Fig. RL-9 is included in SI as Supplementary Fig. 21-26.

Page 11, Lines 3-19:

‘In addition, we conducted additional experiment to confirm the structural stability of Au₂₅ NCs during the electrocatalytic reactions. We performed UV-vis absorption spectroscopy and ESI-MS to determine the formula of Au NCs before and after oxygen evolution reaction (OER). The UV-vis absorption spectra before and after 200 cycles of LSV tests show almost identical profiles with minor differences in the intensity, where slight decrease in absorption intensity was observed at ~695 nm (Supplementary Fig. 21). This reflects that the majority of [Au₂₅(pMBA)₁₈]⁻ survive after the OER although a trivial degradation is not avoidable. The ESI-MS of Au₂₅ NCs capped by pMBA after OER process further confirm their unchanged formulae after OER (Supplementary Fig. 22). With respect to [Au₂₅(MHA)₁₈]⁻ and [Au₂₅(HCys)₁₈]⁻ NCs, more distinct decrease in absorption peak intensity was observed. (Supplementary Fig. 23 and 24), indicating that they are less stable than [Au₂₅(pMBA)₁₈]⁻. Nevertheless, their formulae are [Au₂₅(MHA)₁₈]⁰ and [Au₂₅(HCys)₁₈]⁰ NCs, which are unearthed by ESI-MS (Supplementary Fig. 25 and 26). Overall, the atomic-level morphology is not changed for Au₂₅ NCs capped by these three ligands before and after the OER test.’

Page 14, Fig. 5:

Fig. RL-10 is included in the main text as Fig. 5.

Supplementary Information (SI), Page 31:

Fig. RL-11 is included in SI as Supplementary Fig. 28.

Page 14, Lines 13-19 and Page 15, Lines 1-4:

‘To understand the ligand effect on the OER performance of Au₂₅ NCs in alkaline conditions, the adsorption energies of OH and O on Au₂₅ NCs were calculated to compare pMBA and MHA ligands. To save the computational cost, we employed the models of [Au₂₅(SCH₃)₁₅(SR)₃]³⁻ (SR = pMBA or MHA) where three deprotonated real thiolate ligands are present on the same dimeric motif while other thiolate ligands are simplified as SCH₃ (more computational details can be found in the section Methods). As shown in supplementary Fig. 28 and 29, both O and OH adsorb on an Au(I) atom in the dimeric motif and, more importantly, their adsorption is less stronger with the presence of the pMBA ligand than the MHA ligand,

indicating that *OH and *O species are easier to be activated on [Au₂₅(pMBA)₁₈]⁻. Such correlations of adsorption energy with OER activities have been observed by others as well^{48,49}.’ (*Nat Catal.* **2018**, 1, 300–305; *ACS Catal.* **2022**, 12, 11597–11605)

2.The effect of OH/ligand exchange on the electrochemical data should also be discussed.

Reply: Thanks for your suggestions. We have excluded the likelihood of extensive OH/ligand exchange in OER. As discussed above, the UV-vis absorption and ESI-MS analysis confirmed the structural stability of the [Au₂₅(SR)₁₈]⁻ NCs after OER processes. It should be pointed out that the catalytic activity of [Au₂₅(SR)₁₈]⁻ NCs does not necessarily rely on the desorption of SR ligand. This is because Au atom in the protecting motifs is under-coordinated, which can serve as open active sites for OER without ligand exchange. Similar observations of catalytic activity on ligand-on Au NCs have been extensively documented for Au NCs elsewhere (*Chem. Rev.* **2021**, 121, 567–648).

3.A question regarding Figure 3 / text page 8: Where do the over potential values (line 193, the 540 mV supposedly belong to the MHA-ligand cluster? typo twice HCys?) stem from? How were they extracted from the Figure (3a? or which?)? In Figure 3a, the MHA-10 mA/cm² is hardly in the figure anymore, it seems. A discussion on how these overpotential values compare to literature should be included for the reader to understand the quantitative (potential) impact of ligand tuning on the reactivity. What are the pure (ligand-free) Au overpotential values?

Reply: We are sorry for the typos as noted by the reviewers. We have corrected these typos in the revised manuscript. The overpotentials are estimated by comparing the potential at 10 mA/cm² (can be readable from LSV curve in Fig. 3a) with the theoretical potential value (1.23 V) for water splitting. To make it more readable, we have added a reference dotted line in Fig. RL-13 (also as Fig. 3a), indicating the current density of 10 mA/cm². It’s clearer to read the applied potential value (i.e., 1.77 V vs. RHE) for

$[\text{Au}_{25}(\text{MHA})_{18}]^-$ at 10 mA/cm^2 . Therefore, the overpotential (η) for $[\text{Au}_{25}(\text{MHA})_{18}]^-$ is, $\eta = 1.77 - 1.23 \text{ V} = 0.54 \text{ V}$.

Fig. RL-13 (also as Fig. 3a) Linear sweep voltammetry (LSV) curves of different-ligands-capped $[\text{Au}_{25}(\text{SR})_{18}]^-$ NCs for OER recorded at a scan rate of 5 mV s^{-1} after iR-corrected.

Moreover, we also compared our Au_{25} NCs with other transition metal electrocatalysts protected by thiolate ligand in terms of the overpotential required for reaching a current density of 10 mA/cm^2 (Fig. RL-14, also as Supplementary Fig. 10). As can be seen, $[\text{Au}_{25}(\text{pMBA})_{18}]^-$ NCs show a better performance, while $[\text{Au}_{25}(\text{HCys})_{18}]^-$ and $[\text{Au}_{25}(\text{MHA})_{18}]^-$ are comparable to other transition metal based electrocatalysts.

Fig. RL-14 (also as Supplementary Fig. 10) Comparison of overpotentials required for reaching a current density of 10 mA/cm². (Ni₆(2-phenylethanethiol)₁₂, Ni₅(2-phenylethanethiol)₁₀, and Ni₄(2-phenylethanethiol)₈ (*Inorg. Chem.* **2023**, 62, 1875–1884); Pd₆(SC₁₂H₂₆S)₁₂ (*Chem. Commun.* **2017**, 53, 9733–9736); Au₂₅(PET)₁₈ (*Nanoscale*, **2020**, 12, 9969–9979); Copper (I)-thiolate cluster (*Chem. Commun.* **2020**, 56, 3967–3970), Ni₆(phenylethyl)₁₂ (*ACS Catal.*, **2016**, 6, 1225–1234); The overpotentials in this work are marked in stars filled with different colors.

As for ligand-free Au NPs, we have performed additional experiment according to the reviewer's advice. We compared the electrocatalytic performance of [Au₂₅(MHA)₁₈]⁻ with that of ligand-free Au nanoparticles (NPs, ~100 nm in diameter, prepared by NaBH₄ reduction). The LSV curves in the Fig. RL-15 below indicates that the highest current density delivered by Au NPs within the potential window from 1.1 to 1.8 V is only up to ~6 mA/cm², which is lower than the current density delivered by [Au₂₅(MHA)₁₈]⁻ NCs in the same potential range. Of note, [Au₂₅(MHA)₁₈]⁻ NCs perform the worst among the examined three Au₂₅ NCs in the aforementioned potential range.

Fig. RL-15 LSV curve comparison for [Au₂₅(SR)₁₈]⁻ NCs and ligand-free Au NPs.

Revisions:

Page 8, Fig. 3a:

Fig. RL-13 is included in the main text in Fig. 3a.

Supplementary Information (SI), Page 13, Supplementary Fig. 10:

Fig. RL-14 is included in SI as Supplementary Fig. 10.

Page 9, Lines 10-15:

‘Moreover, we also compared our Au₂₅ NCs with other thiolate-protected transition metal electrocatalysts in terms of the overpotential required for reaching a current density of 10 mA/cm² (Supplementary Fig. 10). As can be seen, [Au₂₅(pMBA)₁₈]⁻ NCs display a better performance, while [Au₂₅(HCys)₁₈]⁻ and [Au₂₅(MHA)₁₈]⁻ are comparable to other transition metal based electrocatalysts.’

4. What are the error bars of the generated O₂ volume data points in Figure 3e?

Reply: Many thanks to the reviewer’s insightful comments. We have added the error bars accordingly.

Revisions:

Page 8, Fig. 3e:

Fig. 3e O₂ production volumes as a function of water-splitting time by using [Au₂₅(pMBA)₁₈]⁻ as electrocatalysts: the circles are experimentally obtained O₂ volumes, while the solid line indicates theoretical value of O₂ calculated by assuming a 100% Faradaic efficiency for the anode reaction at the current density of 10 mA/cm². The error bars correspond to one standard deviation.

5. What are the applied potentials in Figure 3f?

Reply: We gratefully appreciate the reviewer's careful review. We have marked all the applied potentials (i.e., 1.57, 1.62, 1.66 and 1.59 V vs. RHE) in Fig. 3f in the revised manuscript.

Revision:

Page 8, Fig. 3f:

Fig. 3f I-t curves of $[\text{Au}_{25}(\text{pMBA})_{18}]^{-}$ NCs with different applied electrode potentials. The electrocatalytic reactions were carried out in O_2 -saturated aqueous solution of 1 M KOH.

6.Regarding the Tafel slopes, the discussion on page 9 should be more detailed. In line 222, the authors state that the slopes for MHA and HCys ligands are 'a little higher' than for pMBA ligands. In fact, they are a factor 3 and 5 higher, respectively, which is quite a lot. What do these values mean? How do the authors deduce in such a straightforward way the intermediates that form during the reaction, given the strong electronic ligand effect? What does the sentence 'additional overpotentials ... were required to stabilise Au-OH for adequate coverage'?

Reply: Thanks for your comments. We have made a major revision on the Tafel analysis in page 9 of the revised manuscript. In general, Tafel analysis is utilized to elucidate the reaction mechanism and compare the electrocatalytic activity. For Tafel analysis, the sensitivity (i.e., Tafel slope) of the electric current response to the applied potentials could provide information associated with the rate-determining step. Different Tafel slopes indicate different RDS for elementary step involved in OER.

This can be accomplished by comparing the experimentally observed slopes with those theoretically derived slopes by assuming different rate-determining step based on the well-developed thermodynamic-kinetic model in the past decades (*ACS Catal.* **2020**, 10, 8597–8610; *J. Am. Chem. Soc.* **2019**, 141, 13803–13811; *ACS Catal.* **2014**, 4, 4364–4376; *Chem. Rev.* **1996**, 96, 3177–3200). We had revised the corresponding discussion in page 9. To avoid any misleading, we have also deleted the ambiguous statement, which reads ‘additional overpotentials were required to stabilise Au-OH for adequate coverage.’

Revisions:

Page 9, Lines 26-33 and Page 10, Lines 1-3:

‘As shown in Fig. 3b, the Tafel slope of $[\text{Au}_{25}(\text{pMBA})_{18}]^-$ NCs within the tested overpotential range from 0.32 to 0.36 V is 62 mV/dec, suggesting the decomposition of Au-O-OH might be slower than other elementary steps. In comparison, $[\text{Au}_{25}(\text{MHA})_{18}]^-$ and $[\text{Au}_{25}(\text{HCys})_{18}]^-$ NCs deliver a higher Tafel slope of 304 and 196 mV/dec, respectively. These values imply that the RDS is dominated by deprotonation of Au-OH³⁹. The above-mentioned different Tafel slope values suggest that the RDS is changed by varying the protecting ligands from MHA or HCys to pMBA. The larger Tafel slope reflects stronger polarization as the current density rises. Such higher Tafel slope value presumably has a variety of causes. On the one hand, Tafel slope varies depending on different RDS. On the other hand, the higher activation energy on intermediates possibly makes Tafel slope value higher⁴⁰.’

7. In the discussion of the mechanism, p13, line 315ff, the authors discuss a ‘threshold [OH] coverage’. What does this mean? Why is it needed? Why not correlate this statement with the Raman data? How do the authors distinguish Au-O, Au-OH and Au-OOH species? How is the ligand-O(H) place exchange entangled with the coverage discussion?

Reply: We thank the reviewer for pointing out this issue. We have revised accordingly and combined this discussion with Raman data according to the reviewer’s advice. It has been widely known that the OER current depends on electrocatalytic reaction rate. The reaction rate is closely associated with the coverage of intermediate species (*J.*

Electrochem. Soc. **2017**, 164 E3321; *Angew. Chem. Int. Ed.* **2023**, 62, e202300054). Therefore, the coverage for the intermediates will be affected by the applied potential. There is a threshold value for the intermediate coverage to change the rate-determining step (RDS) (*Sci. Rep.* **2015**, 5, 13801). Specifically, if an intermediate participates in the RDS as reactant, it will accumulate on the surface of metal NCs, boosting its surface coverage. This also provide an opportunity for us to capture the intermediates using *in situ* Raman spectroscopy. We plotted the intensity against applied potential in Fig. RL-16 (also as Fig. 4d). Although the peak positions vary with the protecting ligand changed from aromatic (i.e., pMBA) to aliphatic (i.e., MHA and HCys), the peak intensity shows a ubiquitous ascending trend with the increase of the applied potentials regardless of the capping agent used. This suggests that the coverage of intermediate species, either Au-OH or Au-O-OH, increases with the elevating overpotential, supportive to the proposed electrocatalytic mechanism.

Fig. RL-16 (also as Fig. 4d) Plot of the Raman intensity vs. the increasing potentials.

Regarding the assignment of intermediates, the reported experimental and theoretical vibration values for the Au-O, Au-OH, and Au-O-OH species were used to distinguish these three species. First, assignment of the observed peak in our experiment situated at ~ 600 and 700 cm^{-1} to $\nu(\text{Au-OH})$ as opposed to $\nu(\text{Au-O})$ is supported by the following observations: a recent study affirmed the Au-OH stretching vibration located at 635 and 677 cm^{-1} , whereas a feature located in the range of $373 \sim 568\text{ cm}^{-1}$ belongs to $\nu(\text{Au-O})$. (*ChemPhysChem* **2010**, 11, 1854–1857; *ACS Catal.* **2020**, 10, 12582–12589; *Angew. Chem., Int. Ed.* **2017**, 56, 12952–12957). As for Au-OOH, the signals in the range of $\sim 810\text{ cm}^{-1}$ were assigned to Au-O-OH stretching based on a

series of DFT calculations (*Chem. Phys.* **2005**, 319, 178–184). Moreover, the peroxy species (Au-OO) were excluded due to its extraordinary short lifetime (*J. Am. Chem. Soc.* **2014**, 136, 10432–10439; *Angew. Chem., Int. Ed.* **2017**, 56, 12952–12957).

We have excluded the likelihood of extensive OH/ligand exchange in OER. As discussed above, the UV-vis absorption and ESI-MS analyses confirmed the structural stability of the $[\text{Au}_{25}(\text{SR})_{18}]^-$ NCs after the OER processes. It should be pointed out that the catalytic activity of $[\text{Au}_{25}(\text{SR})_{18}]^-$ NCs does not necessarily rely on the desorption of SR ligand. This is because Au atom in the protecting motifs is under-coordinated, which can serve as active sites even with ligand coordinated. Similar observation of catalytic activity on ligand-on Au NCs have been extensively documented (*Chem. Rev.* **2021**, 121, 567–648).

Revisions:

Page 13, Lines 8-15:

‘It has been widely accepted that OER current depends on electrocatalytic reaction rate. The reaction rate is closely associated with the coverage of intermediate species. Therefore, the coverage for the intermediates will be affected by the applied potential, and a threshold value is required for a specific intermediate coverage to trigger its extensive conversion⁴⁶. Specifically, if an intermediate specie participates in the RDS as reactant, it will accumulate on the surface of metal NCs, boosting its surface coverage. This also provide an opportunity for us to capture the intermediates using *in situ* Raman spectroscopy.’

8.Last paragraph should probably be called ‘summary’ rather than discussion.

Reply: We totally agree with the reviewer. We have revised the section title accordingly.

Revisions:

Page 15, Line 16:

‘Summary’

Comments by Reviewer #3:

The authors reported the ligand effect of atomically precise gold nanoclusters (NCs) on electrocatalytic OER kinetics. Three well-defined model Au₂₅ NCs capped by MHA, HCys, and pMBA were studied. The Au₂₅(pMBA)₁₈ exhibited the highest OER activity among the three NCs, which was ascribed to more partial positive charge on Au(I) induced by pMBA with a stronger electron-withdrawing ability. Different RDSs were suggested based on the Tafel analysis and in-situ Raman spectroscopy. This is an interesting result showing the ligand effect on the electrocatalysis RDS, but some evidence needs to be strengthened to support the conclusions. The manuscript is recommended for consideration after addressing the following concerns.

Reply: We are excited about the reviewer's positive comments on the novelty and significance of our manuscript. We believe the correlations between the molecular-level structure and performance of metal nanoclusters (NCs) may provide an in-depth understanding on design principles of nanomaterials for effective electrocatalysis. Our investigations will fill in the gap between the fundamental chemistry of metal NCs and application-oriented design methodology of metal nanomaterials. We have taken the detailed and useful comments of the reviewer into careful consideration in this revision. Below is an account of our point-to-point response to the reviewer's specific comments/suggestions.

1. The authors suggested different RDS of OER for Au₂₅ NCs capped by MHA, HCys, and pMBA based on the in-site Raman spectra. What is the rationale to connect the reaction intermediate with the reaction RDS? Observation of reaction intermediate does not necessary indicate the reaction RDS.

Reply: Thanks for your insightful comments. Intermediate specie usually participates in specific elementary step reaction of OER. If an intermediate species participates in the RDS as reactant, it will accumulate on the surface of metal NCs. This offers an opportunity for us to capture such species by *in situ* Raman spectroscopy. In addition to Raman spectroscopy, the RDS is also inferred by the Tafel analysis. Tafel analysis is usually utilized to elucidate the reaction mechanism and compare the electrocatalytic

activity. For Tafel analysis, the sensitivity (i.e., Tafel slope) of the electric current response to the applied potentials could provide information associated with the RDS. Different Tafel slopes indicate different RDS for elementary step involved in OER. Similar Tafel analysis derived mechanistic explorations have been extensively reported in the literatures (*ACS Catal.* **2020**, 10, 9271–9275; *Angew. Chem. Int. Ed.* **2019**, 58, 12252–12257; *J. Am. Chem. Soc.* **2020**, 142, 11901–11914). Therefore, the combined experimental (*in situ* Raman spectroscopy and Tafel slope analysis) analysis consolidate the proposed OER mechanism, where RDS is alterable vis surface engineering of Au₂₅ NCs.

Revisions:

Page 13, Lines 20-23:

‘Combined with Tafel slope analysis on different RDS for Au₂₅ NCs capped by different ligands, the Raman data provide molecular-level information on the intermediate species accumulated on the surface of Au₂₅ NCs, hinting on the RDS of OER.’

2. *The thiolate-protected Au NCs may undergo dethiolation process during electrocatalytic reaction. Is the integrity of Au₂₅(pMBA)₁₈ preserved after electrocatalytic reaction?*

Reply: This is an excellent question. We have conducted additional experiment to clarify if [Au₂₅(SR)₁₈][−] maintain the integrity before and after electrocatalytic reaction. We performed UV-vis absorption spectroscopy and ESI-MS to determine the formula of Au NCs before and after oxygen evolution reaction (OER). The UV-vis spectra before and after 200 cycles of LSV tests show almost identical profiles with minor differences in the intensity, where slight decrease in absorption intensity was observed at ~ 695 nm (Fig. RL-17a also as Supplementary Fig. 21). This reflects that the majority of [Au₂₅(pMBA)₁₈][−] survive after the OER although a trivial degradation is not avoidable. The ESI-MS of Au₂₅NCs capped by pMBA after OER process further confirm their unchanged formulae after OER (Fig. RL-17b also as Supplementary Fig. 22). With respect to [Au₂₅(MHA)₁₈][−] and [Au₂₅(HCys)₁₈][−] NCs, more distinct decrease

in absorption peak intensity was observed. (Fig. RL-17c and 17e also as Supplementary Fig. 23 and 24), indicating that they are less stable than $[\text{Au}_{25}(\text{pMBA})_{18}]^{-}$. Nevertheless, their formulae are $[\text{Au}_{25}(\text{MHA})_{18}]^0$ and $[\text{Au}_{25}(\text{HCys})_{18}]^0$ NCs, which are unearthed by ESI-MS (Fig. RL-17d and 17f also as Supplementary Fig. 25 and 26). Overall, the capping ligands on Au_{25} NCs within this study were retained before and after the OER tests.

Fig. RL-17 (a, c, e) UV-vis absorption spectra; (b, d, f) ESI-MS of $[\text{Au}_{25}(\text{SR})_{18}]^{-}$ before and after homogenous OER. The filled curves in the insets are simulated isotope patterns of labelled cluster formulae.

3. The charge state of Au_{25} NCs should be carefully determined in comparison with their open-circuit potential corresponding to the anionic form, i.e., $\text{Au}_{25}(-1)$. Please indicate the open-circuit potentials on the cyclic voltammograms and determine the charge state with respect to the open-circuit potential.

Reply: Thanks for your helpful suggestions. We have marked the position of open-circuit potentials in the Fig. RL-18 (Supplementary Fig. 11-13). Compared with the onset potentials (~ 1.5 V can be readable from LSV curve in Fig. 3a) of synthesized Au_{25} NCs, the observed oxidation peaks for these Au_{25} NCs are assigned more positive

potential with a reference to the open-circuit potential, indicating $[\text{Au}_{25}(\text{SR})_{18}]^-$ evolves into $[\text{Au}_{25}(\text{SR})_{18}]^0$ before OER.

Fig. RL-18 (also as Fig. Supplementary 11-13) cyclic voltammograms (CV) curves of $[\text{Au}_{25}(\text{SR})_{18}]^-$ NCs (a) $[\text{Au}_{25}(\text{MHA})_{18}]^-$; (b) $[\text{Au}_{25}(\text{HCys})_{18}]^-$; (c) $[\text{Au}_{25}(\text{pMBA})_{18}]^-$.

Revisions:

Supplementary Information (SI), Page 14-16, Supplementary Fig. 11-13:

Fig. RL-18 is included in SI as Supplementary Fig. 11-13.

4. On Page 12, line 277: “the quantum size effect” is quite ambiguous. How does the quantum size effect affect the Raman shift?

Reply: Thanks for your insightful comment. We are sorry for the ambiguity. In fact, we intended the ‘size effect’ rather than ‘quantum size effect’ on Raman shift. In general, Raman spectroscopy displays a size-dependent Raman shift behavior (*Sci. Rep.* **2016**, 6, 20539). It was well established that Raman signal was less distinct in low-dimensional nanoparticles when compared to the corresponding bulk crystal spectra (*Angew. Chem. Int. Ed.* **2017**, 56, 12952–12957; *ChemPhysChem* **2010**, 11, 1854–1857). In bulk materials or large nanoparticles, the vibrational modes are well defined and narrow due to the free propagating space (*J. Raman Spectrosc.* **2007**, 38, 618–633). However, due to the high surface-to-volume ratio in nanocluster, the phonons are confined to a small volume, whose size is comparable to the wavelengths of phonons. As a result, the vibrational modes become broader and more complex. We have corrected this terminological mistake.

REVIEWERS' COMMENTS

Reviewer #1 (Remarks to the Author):

The authors have put significant effort into addressing the comments, and have addressed them all to my satisfaction. I support publication in the current form.

Reviewer #2 (Remarks to the Author):

All my concerns and questions have been addressed with great care and detail; I think the same holds for the other reviewers' comments. As such, I recommend publication of the manuscript in the revised form (there are still some typos in the text that should be corrected).

Reviewer #3 (Remarks to the Author):

In the revised manuscript, my previous concerns have been addressed satisfactorily. I recommend this article for publication in Nature Communications.

Replies to reviewers' comments and descriptions of revisions made

Reviewer #1 (Remarks to the Author):

The authors have put significant effort into addressing the comments, and have addressed them all to my satisfaction. I support publication in the current form.

Response: We are glad to learn that the reviewer finds our revisions satisfactory. We would like to thank the reviewer again for your great efforts and helpful suggestions, which have spurred significant improvement in both the scientific content and readability of our manuscript.

Reviewer #2 (Remarks to the Author):

All my concerns and questions have been addressed with great care and detail; I think the same holds for the other reviewers' comments. As such, I recommend publication of the manuscript in the revised form (there are still some typos in the text that should be corrected).

Response: We are thankful to the reviewer for the kind suggestions. We have carefully checked through the text and corrected any typos encountered. Thank you for your insightful comments and sharp eyes again.

Reviewer #3 (Remarks to the Author):

In the revised manuscript, my previous concerns have been addressed satisfactorily. I recommend this article for publication in Nature Communications.

Response: We appreciate the reviewer's kind recommendation. We are appreciative to the reviewer for evaluating our manuscript and offering helpful comments and suggestions to help us enhance the quality of our manuscript.